# Four dimensions characterize attributions from faces using a representative set of English trait words

Chujun Lin [1✉], Umit Keles[1] & Ralph Adolphs [1,2]

People readily (but often inaccurately) attribute traits to others based on faces. While the details of attributions depend on the language available to describe social traits, psychological theories argue that two or three dimensions (such as valence and dominance) summarize social trait attributions from faces. However, prior work has used only a small number of trait words (12 to 18), limiting conclusions to date. In two large-scale, preregistered studies we ask participants to rate 100 faces (obtained from existing face stimuli sets), using a list of 100 English trait words that we derived using deep neural network analysis of words that have been used by other participants in prior studies to describe faces. In study 1 we find that these attributions are best described by four psychological dimensions, which we interpret as "warmth", "competence", "femininity", and "youth". In study 2 we partially reproduce these four dimensions using the same stimuli among additional participant raters from multiple regions around the world, in both aggregated and individual-level data. These results provide a comprehensive characterization of trait attributions from faces, although we note our conclusions are limited by the scope of our study (in particular we note only white faces and English trait words were included).

[1] Division of the Humanities and Social Sciences, California Institute of Technology, Pasadena, CA, USA. [2] Division of Biology and Biological Engineering, California Institute of Technology, Pasadena, CA, USA. ✉email: clin7@caltech.edu

People attribute a wide range of traits (temporally stable characteristics, see Methods) to other individuals upon viewing their faces, such as demographics (e.g., gender, age), physical appearance (e.g., baby-faced, beautiful), social evaluation (e.g., trustworthy, competent), and personality (e.g., aggressive, sociable)[1,2]. These trait attributions are made ubiquitously and rapidly[3], and are known to influence most subsequent processing, such as conscious perception[4] and memory[5] of the face. Although trait attributions from faces may not reflect people's actual traits and reveal more about our own biases and stereotypes[3,6,7], they can influence social decision-making in real life, ranging from success in job markets and social relationships to political elections and courtroom decisions[8–12].

Despite the considerable amount of work on the topic[3,13–22], it remains unclear how people make these rapid attributions: do they have distinct representations for each of the hundreds of possible words that describe somebody based on the face (which might well vary depending on the language), or do they map their attributions of the face into a much lower-dimensional psychological space? By analogy, we can perceive (and have words for) many different shades of colors but they are all the result of a three-dimensional color space. In the case of color, the answer is easier because we roughly know the biological mechanism for these perceptions (i.e., there are only three kinds of cones in the retina); in the case of trait attributions from faces, the biological mechanism underlying particular perceptions is unclear and we must infer the descriptive dimensions from behavioral data (typically, using participants' ratings of faces on different trait words). Prior approaches have discovered dimensional frameworks that have largely shaped studies both within and outside the field[3,13,15,23–35] but those approaches used only a small number of trait words (typically 12–18) that were common across studies[2,31,36] or in use by lay people[1,23]. Moreover, those words are partly redundant in meaning and may not encompass the full range of trait words that people can use to describe faces. Consequently, the psychological dimensions suggested by such prior studies may be incomplete.

Here we argue that to understand the comprehensive dimensionality of trait attributions from faces, it is essential to investigate a more comprehensively sampled set of trait words. To meet this challenge, we assembled an extensive list of English trait words that people use to describe faces from multiple sources[1–3,8,10,14–20,22,37–39] and applied a data-driven approach with a pretrained neural network to derive a representative subset of 100 traits (Fig. 1a–d). Similarly, we combined multiple extant face databases and applied a data-driven approach with a pretrained neural network to derive a representative subset of 100 neutral face images of white, adult individuals (Fig. 1e–h) [see Methods]. We focus on English words because English is the most-spoken language (native and learned) around the world[40]. We limit our stimulus images to frontal facing, faces of white, adult individuals with what are perceived as neutral facial expressions in an attempt to control for factors, such as racial and age discrimination, which are known to bias face perception[23,41–44]. Relatedly, this restriction of the variance in our face stimuli served to increase statistical power, by eliminating factors that our study did not intend to investigate, such as facial expressions (see Methods). We verified that the 100 selected traits were representative of the trait words English-speaking people spontaneously use to describe the 100 face images (Fig. 2a, b) and that the 100 selected face images were representative of the physical structure of white, adult faces (Fig. 2c, d). We collected ratings of the 100 faces on the 100 traits both sparsely online (Study 1) [750,000 ratings from 1500 participants with repeated ratings for assessing within-subject consistency for every trait] and densely on-site (Study 2) [10,000 ratings from each of 210 participants across North America, Latvia, Peru, the Philippines, India, Kenya, and Gaza]. All experiments were preregistered on the Open Science Framework (see Methods).

## Results

**Broader considerations and study limitations.** Our study is a basic research investigation of the psychological dimensions that people use to make social trait attributions from unfamiliar faces. It offers a specific methodological advance over prior work in this field by representatively sampling study stimuli. This method starts with more comprehensive sets of stimuli, capitalizes on advances in machine learning algorithms to quantify stimuli, and applies statistical procedures to sample stimuli. This method could be flexibly adapted to a wide range of stimuli and other research domains. This methodological improvement discovered four dimensions that differ to some degree from the dimensions discovered in previous work, highlighting the importance of representative stimulus sampling, and suggesting that the psychological space people use to organize social attributions from faces is more complex than previously thought.

The four dimensions we found here describe how people attribute traits to others based on faces—they do not describe people's actual traits. In fact, we cannot make any claims about whether or not these attributions were valid since we do not measure the actual traits of the people whose faces were used as stimuli. It is generally well known that people's trait attributions from faces are not accurate[3] but instead reflect the rater's biases and stereotypes. Indeed, our findings identify four important dimensions that may contribute to biases and stereotypes that people exhibit when viewing faces, which potentially may inform future work on stereotyping.

We attempted to extend and improve on prior work by being more comprehensive in several aspects, including preregistering our studies, representatively sampling our stimuli, analyzing our data with different methods, testing the robustness of our findings against different factors, and replicating our study in different samples and individuals around the world. However, our study also has important limitations, which constrain the generality of our findings.

First, our study is unlikely to be representative with respect to faces in general. We utilized images of front facing, with what would be perceived as a neutral expression, adult, white faces. This decision was made based on three considerations: controlling for factors, such as race and age discrimination, which are known to influence face perception; relatedly, increasing the statistical power for our aims by reducing those sources of variance that fall outside the scope of our study (e.g., facial expressions); and the availability of a sufficient number of faces for representative sampling in extant face databases.

Second, it remains unknown the extent to which our study is representative of the concepts that people use to make social trait attributions. It is possible that a broader range of concepts are commonly used but were not representatively sampled in our study—for instance, those concepts denoted by derogatory or swear words, and slang words. It is also notable that our study focuses on the concepts denoted by words in English—only one of more than 6000 languages that exist today. It is possible that cultures and languages shape the concepts available to make trait attributions from faces, and thus the underlying psychological dimensions. Our Study 2 investigated samples in different regions around the world to test the reproducibility of our findings but it is not intended to survey cultural effects and we make no claims to that effect.

**Fig. 1 Sampling traits a–d and face images e–h to generate a comprehensive set. a** Sampling of traits began by assembling an extensive list of trait words[1–3,8,10,14–20,22,37–39] spanning all-important categories of trait attributions from faces. **b** Each adjective was represented with a vector of 300 semantic features that describe word embeddings and text classification using a state-of-the-art neural network that had been pretrained to assign words to their contexts across 600 billion words[70]. **c** Three filters were applied to remove words with similar meanings, unclear meaning, and infrequent usage (see Methods). **d** The final set of 100 traits consisted of the sampled adjectives and nouns (see Supplementary Table 1). **e** Sampling of face images began by assembling a set of frontal, neutral, white faces from three popular face databases[71–73]. f, Each face was represented with a vector of 128 facial features that are used to classify individual identities using a neural network[74] pretrained to identify individuals across millions of faces of all different aspects and races. **g** Maximum variation sampling[86] was applied to select faces with maximum variability in facial structure in this 128-D space. **h** Multidimensional scaling visualization of the sampled 100 face images (green and orange dots).

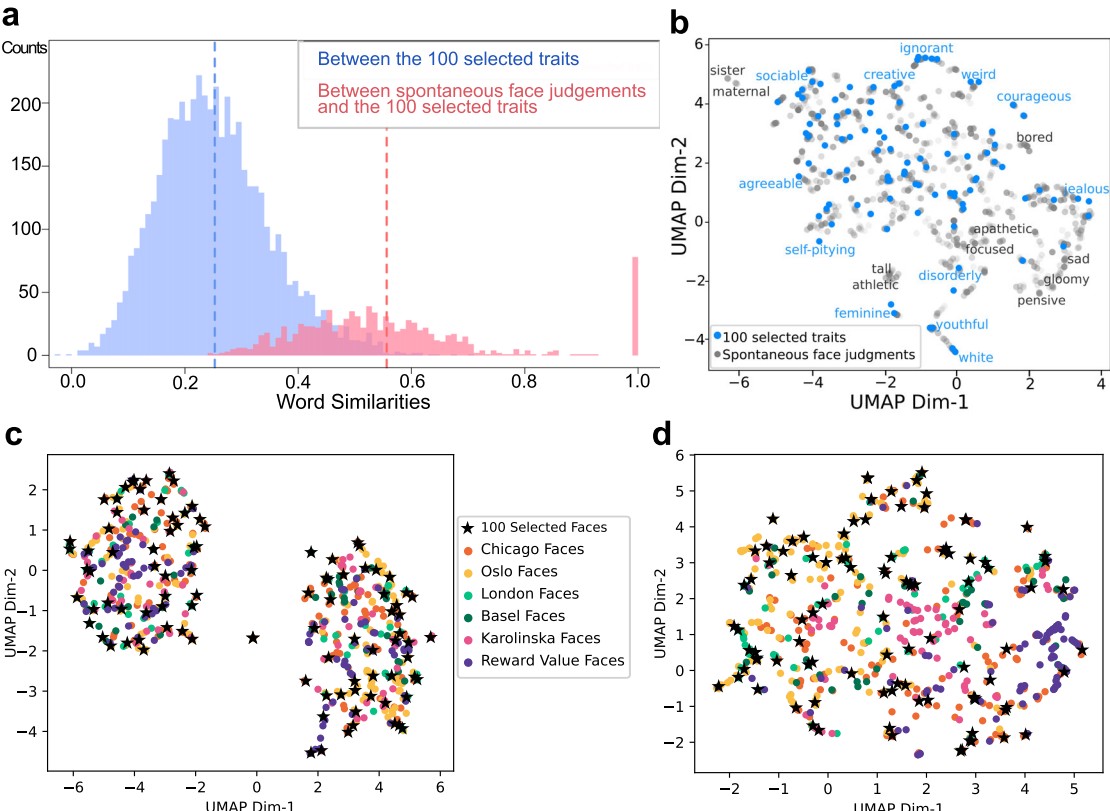

**Fig. 2 Representativeness of the sampled traits a–b and face images c–d. a** Distributions of word similarities. The similarity between two words was assessed with the cosine distance between the 300-feature vectors[70] of the two words. The blue histogram plots the pairwise similarities among the 100 sampled traits. The red histogram plots the similarities between each of the freely generated words during spontaneous face attributions ($n = 973$, see Supplementary Fig. 1a) and its closest counterpart in the sampled 100 traits. Dashed lines indicate means. All freely generated words were found to be similar to at least one of the sampled traits (all similarities greater than the mean similarity among the sampled traits [except for the words "moving" and "round"]). Eighty-five freely generated words were identical to those in the 100 sampled traits. **b** Uniform Manifold Approximation and Projection of words (UMAP[75], a dimensionality reduction technique that generalizes to nonlinearities). Blue dots indicate the 100 sampled traits (examples labeled in blue) and gray dots indicate the freely generated words during spontaneous face attributions (see Methods; nonoverlapping examples labeled in gray, which were mostly momentary mental states rather than temporally stable traits). **c** UMAP of the final sampled 100 faces (stars) compared with a larger set of frontal, neutral, white faces from various databases[76-78] (dots, $N = 632$; see also Supplementary Fig. 1b for comparison with faces in real-world contexts). Each face was represented with 128 facial features as extracted by a state-of-the-art deep neural network[74]. **d** UMAP of the final sampled 100 faces (stars) compared with the larger set of faces (dots) as in **c**. Each face was represented here with 30 automatically measured simple facial metrics[72] (e.g., pupillary distance, eye size, nose length, cheekbone prominence). Source data are provided as a Source Data file.

Finally, our study makes no claims about our four factors being universal, biologically basic, or evolved. This is not only because of the limitations listed above but also because it is unknown what kinds of faces and what kinds of social trait concepts were available to our ancestors.

**Four dimensions underlie trait attributions from faces**. Study 1 examined the underlying dimensions of the ratings that participants had given to the faces (ratings aggregated across participants) by first applying an exploratory method (exploratory factor analysis [EFA]; preregistered) and subsequently a confirmatory method with cross-validation (an autoencoder artificial neural network [ANN]). We confirmed that these ratings showed sufficient variance (Supplementary Fig. 2a), within-subject consistency (assessed with Pearson's correlations, $M = 0.47$, Range = [0.28, 0.84], as well as linear mixed-effect modeling [preregistered]; Fig. 3), and between-subject consensus (preregistered; all ICCs > 0.60) [Fig. 3 and Methods]. Eight traits with low factorizability were excluded from further analyses (Supplementary Fig. 2b; including them did not change the dimensions we eventually found).

We determined the optimal number of factors to retain in EFA using five widely recommended methods[45,46] (see Methods), as solutions are considered most reliable when multiple methods agree. Four methods—Horn's parallel analysis, Cattell's scree test, optimal coordinates, and empirical BIC—all indicated that the optimal number of factors to retain was four (Supplementary Fig. 3a).

EFA was thus applied to extract four factors using the minimal residual method, and the solutions were rotated with oblimin for interpretability. The four factors each explained 31, 31, 11, and 12% of the common variance in the data (85% in total; 87% in total if five factors were extracted) and were weakly correlated ($r_{13} = -0.33$, $r_{14} = -0.23$, $r_{23} = 0.21$, $r_{24} = 0.33$ [$ps = 8.122 \times 10^{-4}$, 0.021, 0.040, $8.358 \times 10^{-4}$]; $r_{12} = -0.15$, $r_{34} = 0.12$ [$ps = 0.129$, 0.237]). None of the factors were biased by words with particularly low or high within-subject consistency or between-subject consensus; and the trait words occupied the four-dimensional space fairly homogeneously (Fig. 3). We interpreted these four factors as describing attributions of warmth, competence, femininity, and youth (Fig. 3; see Supplementary Fig. 4a for factor loadings) [see Methods]. We note that all trait attributions based on faces, and therefore the dimensions describing these attributions, are a reflection of people's

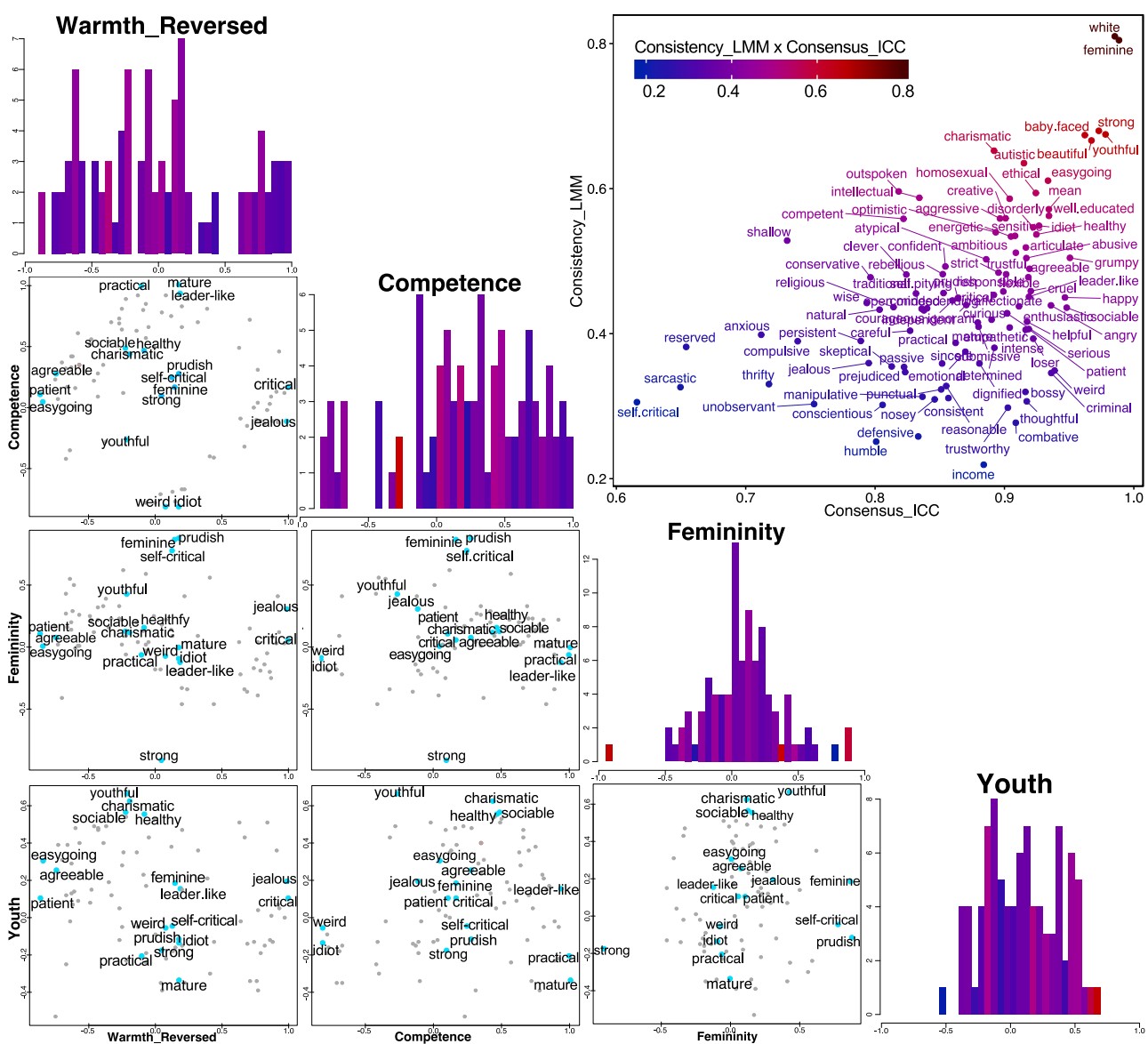

**Fig. 3 Reliability and dimensionality of trait attributions from faces.** Upper right scatterplot: within-subject consistency as assessed with linear mixed-effect modeling (*y*-axis, regression coefficients) plotted against between-subject consensus as assessed with intraclass correlation coefficients (*x*-axis) of the 100 traits. The color scale indicates the product between the *x*- and *y*-values. We used 94 traits selected from the literature and supplemented the list with additional trait words for which we believe there was no equivalent in the initial list but would reflect vocabulary used to describe first impressions. Four histograms in diagonal: each plots the distribution of the factor loadings across all traits in EFA, on each of the four dimensions (color code as in upper right scatterplot; see also Supplementary Fig. 4a for factor loadings). Six scatterplots in the lower left: each plots the factor loading of all traits in EFA against two of the four dimensions (dots). Labels are shown for a small subset of datapoints (blue dots) due to limited space (see Supplementary Fig. 4b for full labels). Source data are provided as a Source Data file.

stereotypes of some sort, since in our study nothing else is known about the people whose faces are used as stimuli. Here we omitted "-stereotypes" in our labeling of all dimensions for conciseness.

To corroborate the four dimensions discovered from EFA, we applied an approach with minimal assumptions—artificial neural networks (ANN) with cross-validation to compare different factor structures (see Methods). Autoencoder ANNs with one hidden layer that differed in the number of neurons (range from 1 to 10) were constructed (Fig. 4a). These ANNs were trained on half of the data (i.e., aggregated ratings across half of the participants) and tested on the other held-out half (Adam optimization algorithm[47] and mean squared error loss function with a batch size of 32 and 1500 epochs were used to train the ANNs, repeated for 50 iterations). Both the linear and nonlinear

activation functions were examined (Fig. 4b). Model performance of the best configuration (i.e., linear activation functions in both the encoder and decoder layers) increased substantially as the number of neurons in the hidden layer increased from 1 to 4 (explained variance on the test data increased by 18, 5, and 5%, respectively); the improvement was trivial beyond four neurons (increased by less than 1%) [Fig. 4c]. Critically, the four-dimensional representation learned by the ANN reproduced the four dimensions discovered from EFA (mean *r*s = 0.98, 0.92, 0.91, 0.94 [SDs = 0.01, 0.05, 0.02, 0.05] between the factor loadings from EFA and the ANN's decoder layer weights with varimax rotation) and confirmed good performance (explained variance obtained with linear activation functions was 75% [SD = 0.6%] on the test data, comparable to PCA).

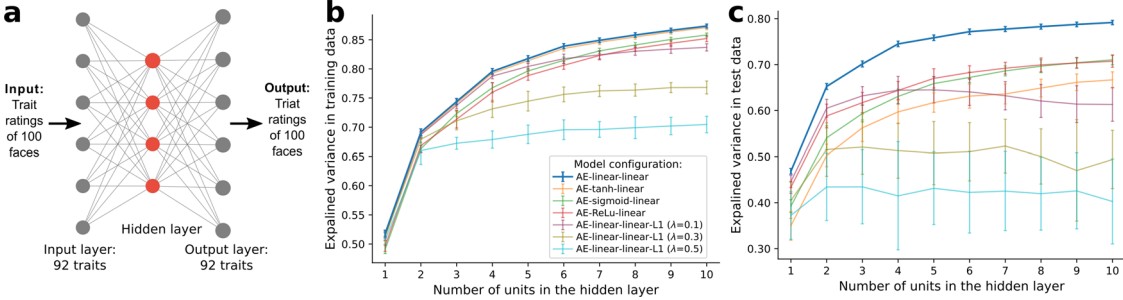

**Fig. 4 Dimensionality analysis with artificial neural network and cross-validation. a** An example of an autoencoder model with one hidden layer and four neurons in the hidden layer to learn the underlying representation of the data. **b**, The means (points) and standard deviations (bars) of the explained variance ($n = 50$ iterations) on the training data from autoencoders with various numbers of neurons in the hidden layer (red dots in a). Colors indicate different configurations of activation functions in the encoder and decoder layers (linear, tanh, sigmoid, rectified linear activation unit, L1-norm regularization); for example, the blue line indicates configurations with linear functions in both the encoder and decoder layers (AE-linear-linear). **c**, Means (points) and standard deviations (bars) of the explained variance ($n = 50$ iterations) on the test data from autoencoders shown in **b**. Source data are provided as a Source Data file.

**Table 1 Factor loadings from EFA on the subset of 13 traits used in the 2D framework.** Factor loadings from EFA on the subset of data corresponding to 13 traits (first column) that are the same or most similar to those used in a prior study that discovered the popular 2D framework[1] (first column, in brackets). Two factors—the optimal number of factors as indicated by both the Cattell's Scree Test and empirical BIC—were extracted and rotated with oblimin. The largest absolute loading across factors for each trait is highlighted in bold. Source data are provided as a Source Data file.

| Traits from our set [traits in 2D framework[1]] | Valence | Dominance |
|---|---|---|
| Sociable [Sociable] | **0.89** | 0.14 |
| Weird [Weird] | **−0.88** | 0.13 |
| Beautiful [Attractive] | **0.86** | 0.03 |
| Confident [Confident] | **0.85** | −0.53 |
| Responsible [Responsible] | **0.82** | 0.12 |
| Trustworthy [Trustworthy] | **0.77** | 0.38 |
| Wise [Intelligent] | **0.70** | −0.06 |
| Thoughtful [Caring] | **0.64** | 0.55 |
| Happy [Unhappy] | **0.54** | 0.45 |
| Submissive [Dominant] | −0.18 | **1.00** |
| Aggressive [Aggressive] | −0.13 | **−0.90** |
| Mean [Mean] | −0.22 | **−0.86** |
| Emotional [Emotionally stable] | 0.48 | **0.54** |

**Comparison with existing dimensional frameworks**. Prior work[1,2,23,32,36] suggests that attributions from faces with a more limited set of descriptive words can be represented by two or three dimensions. Our findings support the general idea of a low-dimensional space but revealed four dimensions that differ from those previously proposed. One plausible source for this discrepancy could be methodological differences[48,49], which turned out not to be the case: we reanalyzed our data using principal components analysis (PCA), a method used in prior work[1,2,36] in which dimensions are forced to be orthogonal, and reproduced the same four dimensions as reported above (Supplementary Fig. 5a).

Instead, the four-dimensional space did not appear in previous studies because of limited sampling of traits in prior work: we interrogated two subsets of our data which each consisted of 13 traits that corresponded to those used in the discovery of the two most popular prior dimensional frameworks (2D and 3D frameworks[1,36]). The four-dimensional space was not evident when analyses were restricted to these two small subsets of traits;

instead, we reproduced the prior 2D framework (Table 1) and 3D framework (Table 2).

We next showed that using a more comprehensive set of trait words here not only revealed a larger number of dimensions but a dimensional space that is distinct from prior frameworks. While our choice of labels for the first two dimensions (warmth, competence) might suggest correspondence to the two dimensions of the popular prior 2D framework (valence, dominance) due to the semantic similarity between the words, the face attributions these dimensions describe are distinct: using the subset of 13 traits that replicated the 2D framework (Table 1), we found that the warmth dimension and the valence dimension were weakly correlated ($r = 0.41$ based on EFA factor scores; $r = 0.09$ based on scores from PCA, the method used in prior work, with which we also replicated the four dimensions from our full dataset and the 2D framework from the subset of 13 traits); the competence dimension and the dominance dimension were not significantly correlated ($r = 0.01$, $p = 0.894$ based on EFA factor scores; $r = 0.09$, $p = 0.383$ based on PCA scores). We note that the youth dimension found here was highly correlated with the youthful/attractiveness dimension proposed in the prior 3D framework ($r = 0.71$ based on EFA scores; $r = 0.76$ based on PCA scores).

Finally, we directly compared how well different frameworks characterized trait attributions from faces. Using linear combinations of traits with the highest loadings on each dimension as regressors (two for each dimension, due to only two traits loading on one of the dimensions in the 3D framework, Table 2), we found that the four-dimensional framework better explained the variance for 82% of the trait attributions (that were not part of the linear combinations) than did any of the existing frameworks (Supplementary Fig. 5b; mean adjusted $R$-squared across all predictions was 0.81 for the four-dimensional framework, 0.72 for the 3D framework, and 0.72 for the 2D framework).

**Robustness of the four dimensions**. We quantified the robustness of our results both across different numbers of trait words and across different numbers of participants. First, we removed trait words one by one and reperformed EFA to extract four factors as before (all pairs of trait words were ranked from the most to the least similar, and the trait with lower clarity rating was removed from each pair). The four dimensions discovered from the full set versus the subsets of traits were highly correlated (Fig. 5a; see Supplemental Table 2a for the complete list of correlations). Second, we randomly removed participants one by one

**Table 2 Factor loadings from EFA on subsets of 13 traits used in the 3D framework.** Factor loadings from EFA on the subset of data corresponding to 13 traits (first column) that are the same or most similar to those used in a prior study that discovered the popular 3D framework[36] (first column, in brackets). Three factors—the optimal number of factors as indicated by Cattell's Scree Test, the optimal coordinates index, Velicer's MAP test, and empirical BIC—were extracted and rotated with oblimin. The largest absolute loading across factors for each trait is highlighted in bold. Source data are provided as a Source Data file.

| Traits from our set [traits in 3D framework[36]] | Approachability | Youthful/Attractiveness | Dominance |
|---|---|---|---|
| Wise [Intelligent] | **0.92** | −0.37 | 0.02 |
| Trustworthy [Trustworthy] | **0.80** | 0.20 | 0.24 |
| Agreeable [Approachable] | **0.68** | 0.20 | 0.43 |
| Confident [Confident] | **0.63** | 0.13 | −0.63 |
| Happy [No Smile-Big Smile] | **0.61** | 0.21 | 0.26 |
| Beautiful [Attractive] | **0.60** | 0.54 | −0.23 |
| Feminine [Feminine] | **0.31** | 0.28 | 0.20 |
| Youthful [Youthful] | −0.11 | **0.98** | 0.12 |
| Baby-faced [Baby-faced] | −0.09 | **0.82** | 0.31 |
| Healthy [Healthy] | 0.52 | **0.67** | −0.25 |
| White [Pallid-Tanned] | 0.16 | **0.27** | 0.05 |
| Submissive [Dominant] | 0.05 | 0.21 | **0.88** |
| Aggressive [Aggressive] | −0.38 | −0.12 | **−0.79** |

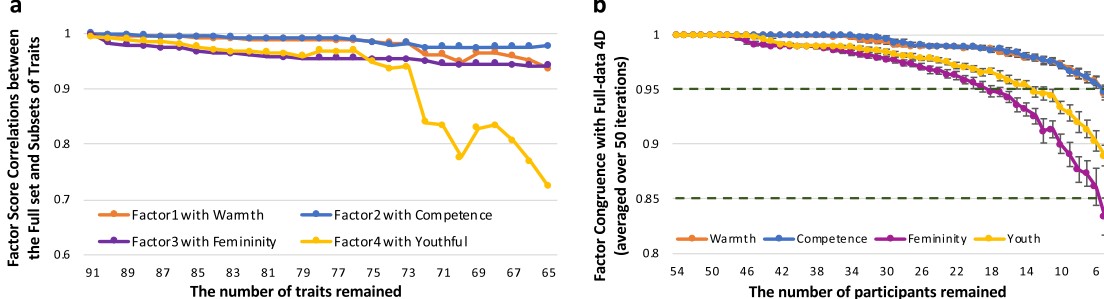

**Fig. 5 Robustness of dimensions with respect to the number of traits and participants. a** Pearson's correlations between factor scores from the full set versus subsets of traits. Colors indicate the four different dimensions. **b** Tucker indices of factor congruence (with orthogonal Procrustes rotation) between the full set versus subsets of participants. Colors indicate the four different dimensions. Points indicate the means and error bars indicate standard deviations across the 50 iterations. Source data are provided as a Source Data file.

(50 randomizations each) and used the new aggregated ratings for EFA to show that the four dimensions discovered from the full dataset were robust to participant sample size (Fig. 5b; Tucker indices of factor congruence >0.95 for all sub-datasets with no fewer than 19 participants per trait).

Finally, we extracted a smaller subset of specific trait words that still yielded the four-dimensional space discovered from the full dataset, a subset of 18 trait words that could be used more efficiently in future studies when collecting ratings for a larger set of traits is not feasible (Supplementary Table 2b). For studies with more stringent constrains on the number of trait words (e.g., due to limited testing time available), an even smaller subset may be selected based on the within-subject consistency and between-subject consensus (Fig. 3; e.g., easygoing, competent, femininity, youthful).

**Results from other countries.** Prior work has reported both the common and discrepant dimensions in different cultures[2,32,48,50,51]. To test the reproducibility of our findings in other subject samples, we conducted a second preregistered study to collect data across seven different regions of the world. We first analyzed the aggregate-level ratings for each sample (preregistered). We confirmed these ratings had satisfactory within-subject consistency and between-subject consensus (see Methods).

We began by asking whether the seven samples shared a similar correlation structure (the Pearson correlation matrix across trait ratings) with the sample in Study 1, using representational similarity analysis[22] [RSA; Fisher z-

transformation was applied before computing the correlation between correlation matrices]. Highly similar correlation structures were found across samples (RSAs with Study 1 = 0.96, 0.92, 0.85, 0.85, 0.75, 0.83, 0.86 for North America, Latvia, Peru, the Philippines, India, Kenya, and Gaza, respectively). These high RSAs strongly suggest that a similar psychological space underlies trait attributions from faces across different samples.

Parallel analysis, optimal coordinates, and empirical BIC all showed that a four-dimensional space was most common across samples (in five of seven samples: North America, Latvia, Peru, the Philippines, India) [Fig. 6a and Supplementary Fig. 3b–h]. We therefore applied EFA to extract four factors from each sample. Results showed that the warmth, competence, femininity, and youth dimensions emerged in multiple samples (interpreted based on factor loadings shown in Supplementary Fig. 6).

We further computed Tucker indices of factor congruence (the cosine distance between pairs of factor loadings), which confirmed that the four-dimensional space was largely reproduced across samples (Fig. 6b); but, as expected, reproducibility was attenuated by within-subject consistency of the data (Fig. 6c)

**Reproducibility across individual participants.** So far, we have reproduced the four-dimensional space across samples but we have not ruled out the possibility that this space might be an artifact of aggregating data across participants. Could the same

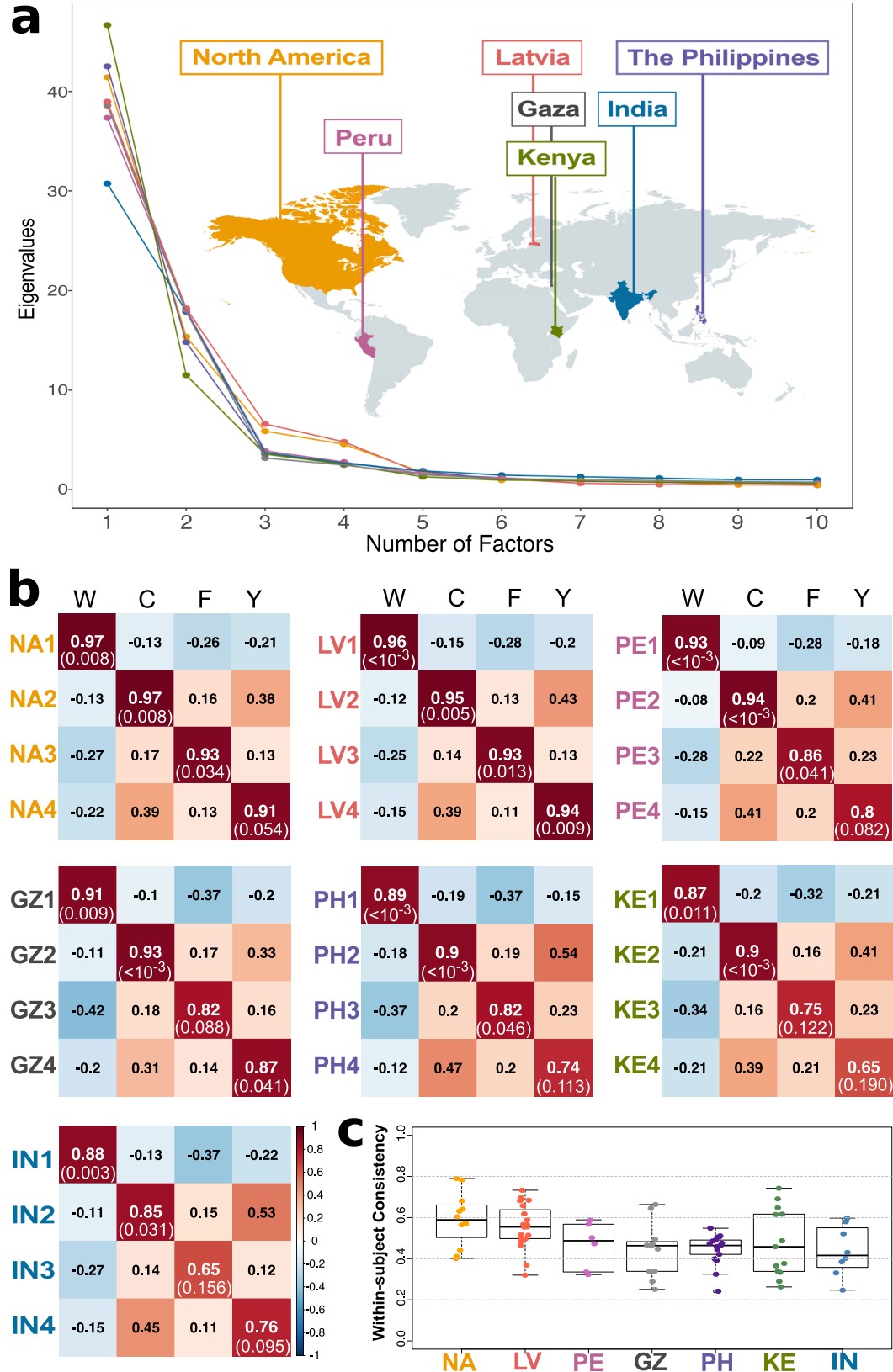

four-dimensional space be reproduced in a single participant? This important question has been difficult to address since one needs to have complete data per participant[52]. We met this challenge by collecting ratings on all traits for all faces from every participant in Study 2 (requiring approximately 10 h of testing per participant; see Methods).

We first performed RSA to investigate whether single participants ($n = 86$ who had complete datasets for all traits after data exclusion; see Methods) shared the correlation structure of our Study 1 sample. RSAs varied considerably across participants (range = [0.14, 0.85], $M = 0.56$, SD = 0.16) and, as expected, were attenuated by within-subject consistency (Fig. 7a, b).

**Fig. 6 Dimensionality of trait attributions from faces across different samples. a** Eigenvalue decomposition. Dots plot the eigenvalues of the first 10 factors across seven samples, indicated by different colors. **b** Tucker indices of factor congruence. Columns indicate the four dimensions found in Study 1: warmth (W), competence (C), femininity (F), and youth (Y). Rows indicate the four factors derived from the samples in North America (NA), Latvia (LV), Peru (PE), the Philippines (PH), Kenya (KE), India (IN), and Gaza (GZ). Numbers report the Tucker indices (with orthogonal Procrustes rotation). The color scale shows the sign and strength of the indices. Statistical significance (p-value in parentheses) was obtained using permutation test (with orthogonal Procrustes rotation, and permuting both the rows and columns of the compared factor loading matrix[87] over 1000 iterations). **c** Individual within-subject consistency by sample (assessed with Pearson's correlations). Every participant in Study 2 had rated a subset of 20 traits twice for all faces to provide an assessment of within-subject consistency (ns = 12, 19, 6, 11, 17, 13, 8 participants from left to right columns who had complete data after exclusion). Boxplots indicate the minima (bottommost line), first quartiles (box bottom), medians (line in box), third quartiles (box top), and maxima (topmost line) of the within-subject consistency. Source data are provided as a Source Data file.

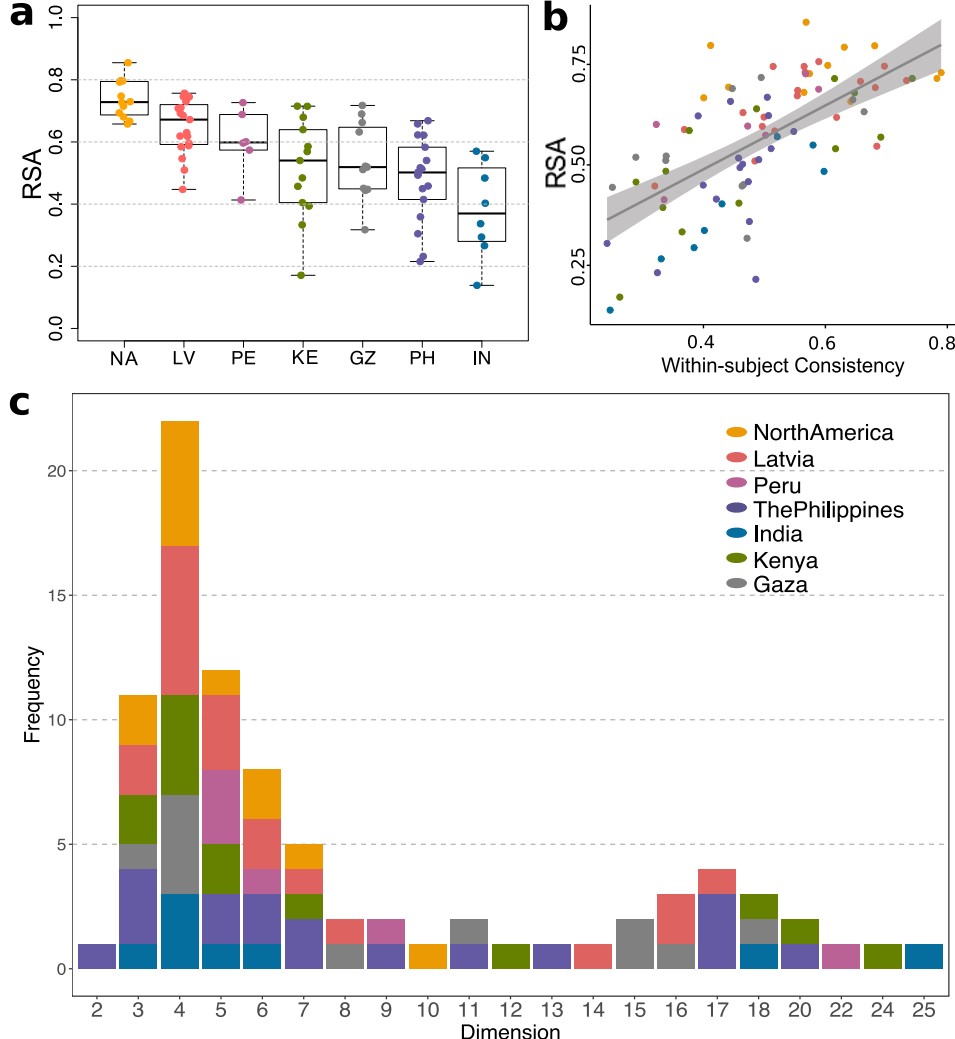

**Fig. 7 Dimensionality of trait attributions from faces in individual data. a** Representational similarity between aggregated data from Study 1 and individual-level data from Study 2 for individuals who had complete data after exclusion (n = 86, see Methods). Colors indicate different samples (as in Fig. 6). Boxplots indicate the minima (bottommost line), first quartiles (box bottom), medians (line in box), third quartiles (box top), and maxima (topmost line) of RSAs. **b** Correlation between within-subject consistency and RSA (r = 0.66, p = 6.476 × 10$^{-12}$). Each point plots an individual's within-subject consistency (x-axis) and that individual's RSA with the aggregated data in Study 1 (y-axis). The shaded area indicates the error band. **c** Distribution of the number of dimensions (from parallel analysis) across 86 individual-level datasets. Source data are provided as a Source Data file.

We next analyzed the dimensionality of each individual dataset. Parallel analysis (preregistered) showed that a four-dimensional space was most common (Fig. 7c) but again, attenuated by within-subject consistency (four-dimensional spaces were found for data with higher within-subject consistency than data that produced other-dimensional spaces [unpaired t-test t(34.57) = 3.29, p = 0.001]). We therefore applied EFA to extract four factors from each participant's dataset and computed their factor congruence with the data from Study 1. We found that the four-dimensional space was successfully reproduced in individual participants (see examples of factor loading matrices in Supplementary Fig. 7a, and Tucker indices for all participants in

Supplementary Fig. 7b) but also found a considerable amount of individual differences, in line with prior research[53].

## Discussion

Across two large-scale, preregistered studies we found that attributions from white adult faces using a more comprehensive set of English trait words than used in previous studies were best described by four psychological dimensions (Figs. 3, 4), interpreted as warmth, competence, femininity, and youth (Supplementary Fig. 4). We showed that our divergence from prior work was not due simply to methodological differences (Supplementary Fig. 5a) but to the prior lack of comprehensively and representatively sampled trait words (Figs. 1, 2 and Tables 1, 2).

We showed that the warmth and competence dimensions reported here captured different aspects of trait attributions from faces than the valence and dominance dimensions previously proposed. However, the relationships between these different dimensions are likely to be modified by the stimuli used[2,23] (see below). Our findings add to those from previous studies[54,55] that attempted to reconcile the dimensions of face perception with the dimensions from the broader social cognition literature, which has long theorized that warmth and competence are two universal dimensions of social cognition[56]. The youth dimension we found here resonates with the youthful/attractiveness dimension found in prior work that used more diverse face images that differed on age[36]. The femininity and youth dimensions are likely linked to overgeneralization[16] and corroborate recent neuroimaging findings on social categorization from face perception[21,57].

This four-dimensional space was reproduced across samples from different regions of the world, even in samples of participants that speak languages other than English [Fig. 6 and Supplementary Fig. 6]. These findings suggest that while languages likely shape the words and concepts available to describe individual trait attribution, the psychological dimensions (which capture the relationship between groups of trait attributions) that people use to represent face attributes are to some extent similar across different countries. This could be tested further in future studies that use no words at all, and instead use measures such as pairwise similarity judgments or pile sorting of the faces[58]. Our four-dimensional space was also reproduced in individual participants (although this was more difficult to assess, due to the heterogeneity in within-subject data consistency) [Fig. 7 and Supplementary Fig. 7].

However, despite the predominance of the four-dimensional space, we also found notable variation across samples and individuals (Figs. 6b, 7c), in line with previous findings[48,52]. Since the sources of this variation are unknown and may largely reflect measurement error (Figs. 6c, 7b), we refrain from drawing any specific conclusions about cultural differences, for which larger-scale studies focusing on cultural effects will be needed[59,60]. Similarly, conclusions about individual differences will require future studies that collect much denser, and likely longitudinal, data in individual participants.

There are several important aspects to note regarding our findings, some of which we elaborate on in the Broader considerations and study limitations in the Results. First, it is critically important to reiterate that our study is one of people's attributions or opinions about faces, and these attributions by people are generally thought to reflect substantial biases and not be valid[3,61]. That is, whether somebody's face is perceived to be competent cannot tell us whether they are actually competent. In fact, many social attributions are fundamentally subject to particular occasions and opinions. Some theories help explain the apparent mismatch between the face attributions that have high consensus, and their invalidity on the other hand. For instance, the overgeneralization theory[16] proposes that many of the attributions people make from faces are a result of overgeneralizing the association between the facial cues that are of evolutionary importance and characteristics of people with those facial cues—for example, the association between facial cues that identify individuals to be babies and that babies are weak and need care, are overgeneralized to the association between the facial cues of "babyfacedness" in adults (e.g., larger eyes, rounder face) and the attribution that those individuals tend to be warm and submissive.

A major limitation of our study is that it is biased with respect to the participants, the trait words, and the faces, a well-known issue with all psychological research historically[62] and also nowadays[63,64]. Although we attempted to improve on prior work in being more comprehensive in these respects, our work was still constrained by the theoretical and practical challenges of sampling participants representative of all people, words representative of all languages, or faces representative of all faces. We elaborate further on these challenges using the case of faces as an example here. Studies such as ours often attempt to derive conclusions about universality: about cognitive features or mechanisms that apply to all people because these processes are in some sense biological and evolved. But this would require the relevant domain of faces to be those faces that our ancestors encountered in the environment in which evolutionary adaptations took place, or perhaps the faces that our participants encountered as children when they learned the culturally stereotypic traits associated with them. These patent difficulties lead us to emphasize once again that our study does not, and cannot, make any claims about human nature or universality.

More specifically, we note that our sampling of trait words was limited to English words; our trait stimuli might thus not be comprehensive for the terms non-English speakers use to describe trait attributions from faces, or might be interpreted differently by people speaking different native languages (even though we provided a one-sentence explanation of each word to help minimize this issue [Supplementary Table 1]). Indeed, it is possible that trait space is more changeable than often thought[65]. While the extent to which our word set might represent the trait concepts of other cultures is unknown, related work on emotion words suggests there may be substantial cultural differences[66]. Finally, the word embedding we applied to quantify and select the trait stimuli also depends on the criteria used by the neural networks (e.g., the corpus of English words used for training, as well as network architecture, loss function), and alternative network parameters might alter the selection outcomes. Future research will be needed to examine how language might modify the psychological space of trait attributions from faces[67,68].

Our sampling of face stimuli was limited to white, unfamiliar, adult individuals; it is possible that features more typical of people of color, or very young or very old individuals, might generate bottom-up modifications to the psychological space, and/or a top-down modulation via the social concepts and stereotypes associated with those populations[2,23,65]. It is also no doubt the case that much of our real-world interaction with people is with those who are familiar, about whom we have extensive additional knowledge, and interactions occur in a rich context. Emotional expressions and contextual factors (e.g., viewing angles, background) will likely further modify the dimensions of trait attributions from faces[3,13,16,17,69]. All of these likely additional sources of variance were precisely the reason that we restricted our stimuli to the more homogeneous set we used in order to reduce the sources of variance in our study: but the result is a constraint on the generality of our findings.

Despite these limitations, we recovered a four-dimensional psychological space that was remarkably resilient to different analysis methods, the number of words used, and the number of participants included, and that showed considerable similarity

across participants in different countries. Our findings provide candidate mental dimensions to investigate in future studies with respect to all the potential modifiers discussed above and emphasize the provisional status that any finding must have in a cumulative science.

## Methods

All studies in this report were approved by the Institutional Review Board of the California Institute of Technology and informed consent was obtained from all participants.

**Sampling of trait words**. Here we follow the definition of a trait as being a temporally stable characteristic. Traits in our study include personality traits as well as other temporally stable characteristics that people spontaneously infer from faces, such as age, gender, race, socioeconomic status, and social evaluative qualities (Supplementary Fig. 1a, e.g., "young," "female," "white," "educated," "trustworthy"). By contrast, we excluded state attributions, such as "smiling" or "thinking" (words that can describe both the trait and state variables were not excluded, e.g., we included "happy," but disambiguated its usage as a trait in our instructions to participants, e.g., "A person who is usually cheerful").

Our goal was to representatively sample a comprehensive list of English trait words that are used to describe people from their faces. We derived a final set of 100 traits (Supplementary Table 1) through a series of combinations and filters (detailed below; also in our preregistration at https://osf.io/6p542). These 100 traits were further verified to be representative of words that people freely generate to describe trait attributions from our face stimuli (Fig. 2a, b).

To derive the final set of trait words, we first gathered an inclusive list of 482 adjectives and six nouns that included all major categories of trait attributions from faces: demographic characteristics, physical appearance, social evaluative qualities, personality, and emotional traits, from multiple sources[1–3,8,10,14–20,22,37–39]. Many of the 482 adjectives were synonyms or antonyms. To avoid redundancy while conserving semantic variability, we sampled these adjectives according to three criteria: their semantic similarity (detailed below), clarity in meaning (from an independent set of 29 MTurk participants), and frequency in usage (detailed below). For those words with similar meanings, clarity was the second selection criterion (the one with the highest clarity was retained). For those with similar meanings and clarity, usage frequency was the third selection criterion (the one with the highest usage frequency was retained).

To quantify the semantic similarity between these 482 adjectives, we represented each of them as a vector of 300 computationally extracted semantic features that describe word embeddings and text classification using a neural network provided within the FastText library;[70] this neural network had been trained on Common Crawl data of 600 billion words to predict the identity of a word given a context. We then applied hierarchical agglomerative clustering (HAC) on the word vectors based on their cosine distances to visualize their semantic similarities. To quantify clarity of meaning, we obtained ratings of clarity from an independent set of participants tested via MTurk ($N = 31$, 17 males, Age ($M = 36$, SD $= 10$)). To quantify usage frequency, we obtained the average monthly Google search frequency for the bigram of each adjective (i.e., the adjective together with the word "person" added after it) using the keyword research tool Keywords Everywhere (https://keywordseverywhere.com/).

The 94 adjectives representatively sampled using the above procedures and the additional six nouns consisted of our final set of 100 trait words. To verify the representativeness of these 100 trait words to words English speakers spontaneously produce, we compared the distributions of our selected words and of 973 words human subjects freely generated to describe their spontaneous impressions of the same faces (see Supplementary Fig. 1a and Methods below), using the 300 computationally extracted semantic dimensions (Fig. 2a, b).

To ensure that the dimensionality of the meanings of the words that we used was not limiting the dimensionality of the four factors we discovered in our study, we derived a similarity matrix among our 100 words using the FastText vector of their meanings in the specific one-sentence definitions we gave to participants in the experiments (Supplementary Table 1; basic stop-words, such as "a," "about," "by," "can," "often," "others" were removed from the one-sentence definitions for the computation of vector representations), and then conducted factor analysis on the similarity matrix. Parallel analysis, Optimal Coordinate Index, and Kaiser's Rule all suggested 13 dimensions; Velicer's MAP suggested 14 dimensions, and empirical BIC suggested five dimensions (empirical BIC penalizes model complexity). We used EFA to extract five and 13 factors using the same method as for the trait ratings (13 factors explained the same common variance as 14 factors, 70%; five factors explained 60%; factors were extracted with minimal residual method and rotated with oblimin to allow for potential factor correlations). None of the dimensions obtained bore resemblance to our four reported dimensions, arguing that the mere semantic similarity structure of our 100 trait words was not a constraint in deriving the four factors that we report.

**Sampling of face images**. Our goal was to derive a set of neutral, frontal, white faces of high quality (clear, direct gaze, frontal, unoccluded, and high resolution)

that are diverse in facial structure. We aimed to maximize variability in facial structure (distinct looking individual faces) while controlling for factors such as race, expression, viewing angle, gaze, and background, which our present project did not intend to investigate and which would reduce statistical power due to additional degrees of freedom. We first combined 909 high-resolution photographs of male and female faces from three publicly available face databases: the Oslo Face Database[71], the Chicago Face Database[72], and the Face Research Lab London Set[73]. We then excluded faces that were not front facing, not with direct-gaze, with glasses or other adornments obscuring the face. We further restricted ourselves to images of white adults and neutral expression. This yielded a set of 426 faces from the three databases.

To reduce the size of the stimulus set while conserving variability in facial structure, we sampled from the 426 faces using maximum variation sampling. For each image, the face region was first detected and cropped using the dlib library[74], and then represented with a vector of 128 computationally extracted facial features for face recognition, using a neural network provided within the dlib library that had been trained to identify individuals across millions of faces of all different aspects and races with very high accuracy[74]. Next, we sampled 50 female faces and 50 male faces that respectively maximized the sum of the Euclidean distances between their face vectors. Specifically, a face image was first randomly selected from the female or male sampling set, and then other images of the same gender were selected so that each new selected image had the farthest Euclidean distance from the previously selected images. We repeated this procedure with 10,000 different initializations and selected the sample with the maximum sum of Euclidean distances. We repeated the whole sampling procedure 50 times to ensure convergence of the final sample. All 100 images in the final sample were high-resolution color images, with the eyes at the same height across images, had a uniform gray background, and were cropped to a standard size. See preregistration at https://osf.io/6p542.

To verify the representativeness of our selected 100 face images, we again performed UMAP analysis[75] to compare the distribution of our selected faces with a) $N = 632$ neutral, frontal, white faces from a broader set of databases[76–78] (Fig. 2c, d) and b) $N = 5376$ white faces in real-world contexts[79,80] that varied in angle, gaze, facial expression, lighting, and backgrounds (Supplementary Fig. 1b), using the 128 computationally extracted facial identity dimensions[74], as well as 30 traditional facial metric dimensions[72].

**Freely generated trait words**. To verify that our selected 100 trait words were indeed representative of the trait attributions English-speaking people spontaneously make from faces, we collected an independent dataset from participants who freely generated words about the person that came to mind upon viewing the face. As preregistered, 30 participants were recruited via MTurk (see preregistration at http://bit.ly/osfpre4); different from the preregistration, we decided to not only include white participants but included participants of any race (27 participants were white, three participants were Black).

Participants viewed the 100 face images one by one, each for 1 s, and typed in the words (preferably single-word adjectives) that came to mind about the person whose face they just saw. Participants could type in as many as ten words and were encouraged to type in at least four words (the number of words entered per trial—words entered by a participant for a face—ranged from zero words [for eight trials] to 10 words [for 190 trials] with mean = five words). There was no time limit; participants clicked "confirm" to move on to the next trial when they finished entering all the words they wanted to enter for the current trial. All data can be accessed at https://osf.io/4mvyt/.

**Study 1 Participants**. We predetermined our sample size for Study 1 based on a recent study that investigated the point of stability for trait attributions from faces:[81] across 24 traits, a stable average rating could be obtained in a sample of 18 to 42 participants (ratings were elicited using a seven-point rating scale, the acceptable corridor of stability was $+/- 0.5$, and the confidence level was 95%). Based on these findings, we preregistered our sample size for Study 1 to be 60 participants for each trait (at https://osf.io/6p542).

Participants were recruited via MTurk ($N = 1,500$ (800 males), Age ($M = 38$ years, SD $= 11$), the median of educational attainment was "some posthigh-school, no bachelor's degree"). All participants were required to be native English speakers located in the US of 18 years old or older, with normal or corrected-to-normal vision, with an educational attainment of high school or above, and with a good MTurk participation history (approval rating ≥ 95%).

We also collected data about whether our participants were currently being treated for a psychiatric or neurological illness. The majority of our participants (79.7%) were not currently being treated for any psychiatric or neurological illness. All dimensional analyses that are reported in the main text on the full sample were repeated also on those 79.7% of participants and the results corroborated all findings from the full dataset: Tucker indices of factor congruence for the four dimensions = 1.00, 1.00, 0.99, 0.99.

**Study 1 Procedures**. All experiments in Study 1 were completed online via MTurk. Considering the large amount of time it would take for a participant to complete ratings for all 100 traits and 100 faces, we divided the experiment into 25 modules:

the 100 traits were randomly shuffled once and divided into 25 modules, each consisting of four traits. Each participant completed one module.

To encourage participants to use the full range of the rating scale, we briefly showed all faces (in five sets of arrays of 20 each) at the beginning of a module, so that participants had a sense of the range of the faces they were going to rate. In each module, participants rated all faces on each of the four traits in the first four blocks (in random order to alleviate carryover effects; we also reanalyzed the data using only the first trait ratings given by participants, and reproduced the four dimensions reported here: Tucker indices of factor congruence = 0.98, 0.97, 0.93, 0.92); in the last (fifth) block they rerated all faces on the trait they were assigned in the first block again, thus providing sparse within-subject consistency data.

At the beginning of each block, participants were instructed on the trait they were asked to evaluate and were provided with a one-sentence definition of the trait (Supplementary Table 1). Participants viewed the faces one by one in random order (each for 1 s) and rated each face on a trait using a seven-point rating scale (by pressing the number keys on the computer keyboard). Participants could enter their ratings as soon as the face appeared or within four seconds after the face disappeared. The orientation of the rating scale in each block was randomized across participants. At the end of the experiment, participants completed a brief questionnaire on demographic information. See preregistration at https://osf.io/6p542.

**Measures of reliability in Study 1.** Data were first processed following three preregistered exclusion criteria (see preregistration at https://osf.io/6p542): of the full sample with a registered size of $N = 1,500$ participants and $L = 750,000$ ratings, $n = 48$ participants, and $l = 27,491$ ratings were excluded from further analysis. Each of the 100 traits was rated twice for all faces by nonoverlapping subsets of participants (ca. $n = 15$ per trait). As preregistered, we applied linear mixed-effect modeling to assess within-subject consistency, which adjusted for non-independence in repeated individual ratings by incorporating both the fixed effects (that were constant across participants) and random effects (that varied across participants). Ratings from every participant for every face collected at the second time were regressed on those collected at the first time (ca. $l = 1,445$ pairs of ratings per trait) while controlling for the random effect of participants.

As preregistered, we assessed the between-subject consensus for each trait with intraclass correlation coefficients (ICC(2,k) using the R function ICC), using ratings of every face by every participant (ca. $n = 58$ participants and $l = 5780$ ratings per trait). A high intraclass correlation coefficient indicates that the total variance in the ratings is mainly explained by the variance across faces instead of participants. We observed excellent between-subject consensus (ICCs greater than 0.75) for 93 of the 100 traits, and good between-subject consensus for the remaining seven traits (ICCs greater than 0.60) [see Fig. 3].

**Determination of the optimal number of factors.** As recommended[45,46,82,83], five methods were included to determine the optimal number of factors to retain in EFA. No single method was regarded as the best method for determining the number of factors; solutions are considered most reliable when multiple methods agree. Parallel analysis retains factors that are not simply due to chance by comparing the eigenvalues of the observed data matrix with those of multiple randomly generated data matrices that match the sample size of the observed data matrix. Prior studies showed that parallel analysis produces accurate estimations of the number of factors consistently across different conditions (e.g., the distribution properties of the data)[82,83]. Cattell's scree test retains factors to the left of the point from which the plotted ordered eigenvalues could be approximated with a straight line (i.e., retains factors "above the elbow"). The optimal coordinates index provides a nongraphical solution to Cattell's scree test based on linear extrapolation. Empirical Bayesian information criterion (eBIC) retains factors that minimize the overall discrepancy between the population's and the model's predicted covariance matrices while penalizing model complexity. Velicer's minimum average partial (MAP) test is "most appropriate when component analysis is employed as an alternative to, or a first-stage solution for, factor analysis"[84]. It is also included in our present study due to its popularity. MAP retains components by partialing out those that resulted in the lowest average squared partial correlation. Parallel analysis, Cattell's scree test, and the optimal coordinates index were computed using $R$ function nScree in the "nFactors" package; eBIC and Velicer's MAP were compute using $R$ function nfactors in the "psych" package.

**Labeling of Dimensions.** Dimensionality reduction methods do not provide labels for the factors discovered, which must instead be interpreted by the investigators. The choice of labels may reflect the biases of the researchers. We note that our third and fourth dimensions describe stereotypes related to gender (femininity-stereotypes) and age (youth-stereotypes) commonly reported in the literature[56]. In fact, essentially all trait attributions based on faces, and therefore all of our dimensions, are a reflection of people's stereotypes of some sort, since in our study nothing else is known about the people whose faces are used as stimuli, and therefore no ground truth is provided. We therefore omitted "-stereotypes" in our labeling of all dimensions, since it implicitly applies to all of them.

**Dimensionality analyses with artificial neural networks and cross-validation.** To compare different theoretical models and test potential nonlinearity in our data, we employed an artificial neural network approach, in particular, autoencoders[85], with cross-validation. The aim of an autoencoder model is to learn a lower-dimensional representation of the data. We constructed different autoencoders based on the different models we wished to test (the existing 2D and 3D frameworks[1,36], the 4D framework from EFA). We trained these autoencoders on half of the data (for each trait, 50% of the individuals were randomly selected and their ratings were used to compute new aggregated ratings per face per trait) and tested them on the held-out other half of the data. We used the Adam optimization algorithm[47] and mean squared error loss function with a batch size of 32 and 1500 epochs to train the neural networks (the loss converged after 1000 epochs in all our models). We repeated this process for 50 iterations and compared the performance of different models. For completeness, both the linear and nonlinear activation functions were explored for model fitting (linear, tanh, sigmoid, rectified linear activation unit, L1-norm regularization, Fig. 4b, c); a simple linear activation function ended up with the best results. These analyses were performed using Keras 2.3.1 with TensorFlow 2.0 in Python 3.6.9.

Existing frameworks[1,36] suggest that all dimensions of trait attributions from faces are of the same order (i.e., no dimension is a higher- or lower-order dimension of the others) but that the number of dimensions varies. Therefore, we first constructed different autoencoder models with only one hidden layer that varied in the number of neurons in this hidden layer, corresponding to the number of underlying dimensions (from 1 to 10). The input layer and output layer were the same for all models, where each face was represented by a vector of ratings across the 92 traits and each trait corresponded to a neuron. All layers were densely connected. We trained these different models and compared their performance (assessed with the explained variance on the held-out test data).

In addition, we tested potential hierarchical factor structure in our data by adding one hidden encoder layer with various numbers of neurons (from 1 to 10) before the middle hidden layer (also with various numbers of neurons from 1 to 10); since autoencoder models are by definition symmetric, these hierarchical latent structures were mirrored in the decoder layers (i.e., three hidden layers). Results showed that adding hidden layers did not increase model performance.

**Study 2 participants.** The study was approved by the Institutional Review Board of the California Institute of Technology and informed consent was obtained from all participants. We preregistered to recruit participants through Digital Divide Data, a social enterprise that delivers research services, in seven countries/regions of the world: North America (U.S. and Canada), Latvia, Peru, the Philippines, India, Kenya, and Gaza. All participants were required to be between 18 and 40 years old, proficient in English (except participants in Peru, where everything was translated to Spanish), have been educated at least through high school, have been trained in basic computer skills, and have never visited or lived in Western-culture countries (except participants in North America and Latvia). In addition, we aimed to have a roughly equal sex ratio of participants in all locations.

The sample size for each location was predetermined to be 30 participants. This sample size was determined based on two criteria: first, the sample size should be large enough to ensure stable average trait ratings (for a corridor of stability of $+ / - 1.00$ and a level of confidence of 95%, the point of stability ranged from 5 to 11 participants across 24 traits[81]); second, the sample size should be feasible to accrue at all seven locations given the requirements mentioned above and the availability of participants for paying multiple visits to complete all our experiment sessions over a 10-day period. See preregistration at http://bit.ly/osfpre2. As planned, 30 individuals (15 females and 15 males) in each of the seven locations participated in our study (Age ($M = 26$, SD = 4) for North America; Age ($M = 28$, SD = 5) for Latvia; Age ($M = 22$, SD = 3) for Peru; Age ($M = 25$, SD = 4) for the Philippines; Age ($M = 27$, SD = 6) for India; Age ($M = 24$, SD = 2) for Kenya; and Age ($M = 26$, SD = 5) for Gaza).

**Study 2 procedures.** All experiments were completed onsite in the Digital Divide Data local offices. Participants in North America, Latvia, the Philippines, India, Kenya, and Gaza completed all experiments in English. Participants in Peru completed all experiments in Spanish. An exact translation of the experiment instructions, trait words, and definitions of the traits from English to Spanish was provided by the Peru office of Digital Divide Data. Both the English and Spanish versions of those materials can be accessed at our preregistration (https://osf.io/qxgmw).

Eighty of the 100 trait words were used in Study 2—twenty words were excluded for low correlations with other traits as found in Study 1 (see Supplementary Figure 2), ambiguity or similarity in meaning as found in feedback from Study 1 (trustful, natural, passive, reasonable, strict, enthusiastic, affectionate, and sincere), and potential offensiveness in some cultures (see Supplementary Figure 6).

Participants in all seven countries/regions followed the same experimental procedures. Each participant provided ratings of all faces on all traits, of which 20 traits were rated twice for within-subject consistency (see our preregistration). The 80 traits were divided into 20 modules, each consisting of four distinct traits (the 20 retested traits were first assigned to distinct modules and then the other traits were randomly assigned across modules with the constraint that traits in the same

module should be balanced in valence). All participants completed all 20 modules during multiple visits to the local offices in ten business days. Each module consisted of five blocks, with the retested trait always shown in the first and last blocks and the other traits shown in random order. The experimental procedure within each module was identical to Study 1.

**Measures of reliability in Study 2.** Data were first processed following our pre-registered exclusion criteria A–C (see preregistration at https://osf.io/tbmsy): of the full sample with a preregistered size of $N = 30$ participants and $L = 300,000$ ratings at each of seven locations ($N = 210$ total), we excluded from further analysis $n = 1$ participant in India and $l = 24,236$ ratings in North America, $l = 2507$ ratings in Latvia, $l = 16,366$ ratings in Peru, $l = 3178$ ratings in the Philippines, $l = 14,389$ ratings in India, $l = 9117$ ratings in Kenya, and $l = 4096$ ratings in Gaza. Registration criterion D was not applied for the analyses of within-subject consistency and between-subject consensus because it imposed a strict lower bound on the within-subject consistency, which might lead to an overestimation of the reliability of the data.

All participants at all locations rated a subset of twenty traits twice for all faces. Analyses of within-subject consistency identical to those in Study 1 were performed for each of the seven datasets ($l = 100$ pairs of ratings across faces per participant for ca. $n = 28$ participants per location). We found acceptable within-subject consistency at all locations ($r_s > 0.20$, except for the ratings of competent, religious, anxious, and critical in India [$r_s = 0.18, 0.18, 0.19, 0.19$] and the ratings of anxious in Peru [$r = 0.19$]). As hypothesized in our preregistration, across all locations, ratings of traits regarding physical appearance had higher within-subject consistency (e.g., feminine, youthful, healthy, with mean $r_s = 0.74, 0.57, 0.51$, respectively) than traits that were more abstract (e.g., critical, anxious, religious, with mean $r_s = 0.31, 0.32, 0.33$, respectively), corroborating findings from Study 1 (Figs. 3, 4).

Assessment of between-subject consensus at each location used data from all participants within the same location ($l = 100$ ratings per participant for the 100 faces from ca. $n = 28$ participants per trait per location). As hypothesized in our preregistration and in line with previous findings[15], traits regarding physical appearance such as feminine, youthful, beautiful, and baby-faced showed high between-subject consensus in all seven locations (all ICCs > 0.86). At the other extreme, some locations had trait ratings with near-zero consensus within that location (the ratings of compulsive in Gaza, prudish in India and Kenya, self-critical in Gaza and the Philippines). This stood in contrast to the findings from Study 1 where ICCs > 0.61 for all the one hundred traits (Fig. 3), and to the samples from North America (ICCs > 0.61 for all traits) and Latvia (ICCs > 0.50 for all traits).

**Data processing for RSA and dimensionality analysis in Study 2.** To ensure high quality and complete data from individuals, we registered four exclusion criteria (A–D) while data collection was underway and data had not yet been analyzed (see registration at https://osf.io/tbmsy), in addition to those planned in our original preregistration (https://osf.io/qxgmw). Analyses of representational similarity and dimensionality for both the aggregated and individual data were performed using data that were processed with exclusion criteria A–D. Following those criteria, thirty-one participants across seven locations were excluded for further analysis ($n = 3$ for North America, $n = 2$ for Latvia, $n = 7$ for Peru, $n = 3$ for the Philippines, $n = 10$ for India, $n = 2$ for Kenya, and $n = 4$ for Gaza). Among those remaining participants, $n = 86$ participants had complete data for all 80 traits —data from these 86 participants were used in the individual-level analyses (Fig. 7).

**Reporting Summary.** Further information on research design is available in the Nature Research Reporting Summary linked to this article.

## Data availability

All de-identified data generated in this study have been deposited in the Open Science Framework: https://osf.io/4mvyt/ and https://osf.io/xeb6w/. Source data are provided with this paper. All face images used in this study are from publicly available databases: https://www.chicagofaces.org/ (Chicago Face Database), https://figshare.com/articles/dataset/Face_Research_Lab_London_Set/5047666 (London Face Database), https://sirileknes.com/oslo-face-database/ (Oslo Face Database). Source data are provided with this paper.

## Code availability

All data were collected via online experiments using custom codes written in Javascript. All data analyses were performed using R (version 3.5.1) and Python (version 3.6.9). All experiment codes and analysis codes are available at Open Science Framework: https://osf.io/4mvyt/ and https://osf.io/xeb6w/.

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

## Acknowledgements

We thank Dean Mobbs, Mark A. Thornton, R. Michael Alvarez, Mark Bowren, Antonio Rangel, Clare Sutherland, Uri Maoz, and William Revelle for their input, Remya Nair and Christopher J. Birtja for technology support, and Becky Santora for helping with testing participants in foreign locations through Digital Divide Data. Funded in part by NSF grants BCS-1840756 and BCS-1845958, the Carver Mead New Adventures Fund, and the Simons Foundation Collaboration on the Global Brain (542941).

## Author contributions

C.L. and R.A. developed the study concept and designed the study; C.L. and U.K. prepared experimental materials; R.A. supervised the experiments and analyses; C.L.

performed and supervised data collection; C.L. and U.K. performed data analyses; C.L. and R.A. drafted the manuscript; all authors revised and reviewed the manuscript and approved the final manuscript for submission.

## Competing interests

The authors declare no competing interests.
