## [Peer Review File · Nature Communications]

Four dimensions characterize attributions from faces using a representative set of English trait wordsREVIEWER COMMENTS

Reviewer #1 (Remarks to the Author):

Human's cognitive system computes high-level attributes from objects, including faces. These attributes are usually assumed to be dimensions of a cognitive or computational space in psychology and neuroscience. Two much-debated dimensions are valence and arousal. While there seems to be quite strong evidence in favor of some valence-line dimension (but see Martinez, 2019, 2018), there's little evidence for the latter. Are there other more appropriate high-level dimensions that define our perception of faces in a social context? The present paper answers this question in the affirmative. Using an innovative approach, the authors examined over 100 different possible attributes and identified four as possible dimensions of the computational space of faces. These results will for sure restart a much-needed debate of the major role of attributes to interpret people's faces.

My only concern with the approach is that the ~100 attributes studied in this paper are selected by the authors, which may lead to bias and a less-than-perfect interpretation of the underlying dimensions of human's cognitive space. For example, we know that languages other than English include affect words not available to English speakers. Thus, using a limited number of dictionary entries limits the results of the paper. A second limitation is in the face datasets used for this study which are highly biased to caucasian/white individuals. I do not see these as disqualifying limitations and believe that the results are important enough to justify publication in a top journal such as this one. However, I'd like to see these limitations noted in the paper. For one this will serve as a cautionary note to readers. And, second, it will hopefully encourage others to extend on this research. Another minor limitation I think the authors should mention is that the word embedding computed with deep neural networks is dependent on the criteria used by that algorithm (e.g., loss function). Again, this is not a major issue, but it should be noted that changing the loss function, architecture, or others, may lead to different outcomes. This one is less important than the other two limitations listed above, mainly because I think the authors have taken good care of this and done the appropriate studies and analyses. Yet, a sentence or two with a cautionary note would go a long way.

In summary, this is a much needed, impressive, highly innovative study with far-reaching results. The paper is well-written. The studies are well designed, and the results are highly relevant to psychologists, neuroscientists, and engineers. I am in strong support for publication.

Some related papers to be considered for additional discussion: Sutherland et al. PNAS 2020, Fan et al., eLife 2020, Srinivasan et al. IEEE TAC 2018, Brooks et al. PNAS 2019, Franklin et al. Emotion 2018.

Reviewer #2 (Remarks to the Author):

SUMMARY OF WORK:

Prior work on social facial perception found that a smaller number dimensions underlie the perception (famously, 2 or 3). Similar to this line of work, the authors determined the number of summary dimensions; they selected face images and trait words that are more representative than those used in the prior work, sampling each of them from a more

comprehensive set of items evenly from the respective space (face space and trait-word space). Then they determined the number of summery dimensions using multiple methods. In sum, this is a strong piece of work that adds to the field. I only have minor issues with the current versions of the manuscript.

PERSONAL EVALUATION OF WORK:

My initial reaction to the paper was mixed. On one hand, I did not find the results presented not too surprising. Sure, employing a bigger, comprehensive set of words and faces will lead to a larger number of underlying dimensions, explaining a bigger variance in data. It seems to me at first this fact makes this work appear incremental in nature. Relatedly, the selection of 100 words and 100 faces seem arbitrary – why not 200, 400, or 1000 face/words? Following the same logic in the introduction, a larger number of stimuli might have led to a larger number of dimensions (although the authors make a convincing cases of the representative of these stimuli, e.g., Fig 2, Supp Fig 1b).

On the other hand, I find the methodological rigor of the current work impressive and the conclusion fair. (1) The authors carefully set each step of analysis, not leaving anything to guess work. They addressed multiple technical issues that were left not considered in prior work, which might have changed the resulting dimensions and the number of the dimensions, albeit potentially to a small degree. I am talking about careful sampling of representative stimuli, preregistrations, confirmatory analyses, robustness and generalizability testing etc. in the present work. Each of these components add to the current paper's value. (2) The figures are helpful and pleasant to look at. (3) Then there are main analyses that are genuinely novel upon a more careful read, e.g., the calculation of each observer's dimensionality and the assessment of cross-observer reproducibility (Study 2).

When everything is taken into consideration, I believe this is a strong piece of work that adds to the field. I commend the authors.

MINOR ISSUES:

1. "By analogy, we can perceive (and have words for) many different shades of colors, but they are all the result of a three-dimensional color space. In the case of color, the answer is easier because we know that there are only three kinds of cones in the retina; in the case of trait judgments of faces, we must infer the psychological space from behavioural data (human subjects' ratings of faces on different trait words)" (p.3)

While this is a compelling analogy, the essential difference between the two systems should be made clear; the cones on the retina have a concrete physical basis, while a n-dimensional trait space does not. The existence of 3 different types of cones explicitly provides the "mechanism" of color perception. On the other hand, n-dimensional trait space (let n be 2, 3, or 4) is simply a statistical summary of behaviour. In other words, the n-d face space is a "description" of a phenomenon (=social facial perception). Based on this, one might even argue that the number of dimensions is unimportant.

2. "true dimensionality" (p.4)

What even is the "true" dimensionality? This is related to my point in MINOR ISSUE No. 1

(the cone and color perception analogy). Personally I am generally uncomfortable with the notion that there are the correct, definite set of underlying dimensions. The dimensions are only a statistical summary of the way we form impressions from faces, influenced by the norm of the time (although according to the paper they seem pretty stable within-observer and cross-culture). I would suggest a rephrase because this like of language ("true") can lead to an essentialist idea and as a result a unfruitful discussion about whether the number of "basic emotion/al expressions" is 4, 5, or 6.

3. "Prior work has reported both common³ and discrepant dimensions in different cultures^{13,14,24,35,37}" (p.15)

The citations should be corrected. For example Ref. no. 13 and 37 only used a English speaking participants in the US and the UK, respectively. As far as I know they did not recruit participants from multiple cultures, and there was no contrast between cultures, which to me seems like a reason you need to remove 13 and 37 from the second list of citations.

4. "but, as expected, reproducibility was attenuated by the data quality available (as assessed by within-subject consistency" (p.16) "as well as in individual participants (although this was more difficult to assess, due to data quality" (p.20) and many other places (e.g., p.18)

Here it seems it is assumed that (1) all regions have relatively same level of within-subject consistency and (2) thus a low within-subject consistency reflects data quality of the sample, NOT a genuine evaluation tendency of the sample. While these are reasonable assumptions, there is no way you can confirm these assumptions given the current data – you do not know the true reason of the low within-subject consistency. Thus the mention of data quality seems inappropriate.

RESPONSE SUMMARY: We thank both reviewers for the helpful and expert critique of our paper. We are grateful for the opportunity to respond to the points raised below.

Reviewer #1 (Remarks to the Author):

Human's cognitive system computes high-level attributes from objects, including faces. These attributes are usually assumed to be dimensions of a cognitive or computational space in psychology and neuroscience. Two much-debated dimensions are valence and arousal. While there seems to be quite strong evidence in favor of some valence-line dimension (but see Martinez, 2019, 2018), there's little evidence for the latter. Are there other more appropriate high-level dimensions that define our perception of faces in a social context? The present paper answers this question in the affirmative. Using an innovative approach, the authors examined over 100 different possible attributes and identified four as possible dimensions of the computational space of faces. These results will for sure restart a much-needed debate of the major role of attributes to interpret people's faces.

RESPONSE: We thank the reviewer for the positive assessment, and for pointing us to two highly relevant papers. We have added those papers as citations in our revised manuscript (lines 14-20, page 21).

My only concern with the approach is that the ~100 attributes studied in this paper are selected by the authors, which may lead to bias and a less-than-perfect interpretation of the underlying dimensions of human's cognitive space. For example, we know that languages other than English include affect words not available to English speakers. Thus, using a limited number of dictionary entries limits the results of the paper. A second limitation is in the face datasets used for this study which are highly biased to caucasian/white individuals. I do not see these are disqualifying limitations and believe that the results are important enough to justify publication in a top journal such as this one. However, I'd like to see these limitations noted in the paper. For one this will serve as a cautionary note to readers. And, second, it will hopefully encourage others to extend on this research. Another minor limitation I think the authors should mention is that the word embedding computed with deep neural networks is dependent on the criteria used by that algorithm (e.g., loss function). Again, this is not a major issue, but it should be noted that changing the loss function, architecture, or others, may lead to different outcomes. This one is less important than the other two limitations listed above, mainly because I think the authors have taken good care of this and done the appropriate studies and analyses. Yet, a sentence or two with a cautionary note would go a long way.

RESPONSE: These are very well-taken and important points. We have acknowledged these limitations now in various places in our revised paper (see below), first and foremost right up-front as the last sentence in the Abstract (page 2, lines 15-17).

In response to the reviewer's first concern, we agree with the reviewer that an unbiased selection of trait attributes is essential for understanding the psychological space of trait

judgments from faces. This is in fact an important motivation of our work: how could one sample trait attributes from more comprehensive sources and select them using more systematic methods than existing approaches? Here we began with trait attributes from English dictionaries—the more than 400 dictionary entries included in our initial trait list were in fact fruitful results from the personality literature, where prior research extracted more than seventeen thousand person attributes from the dictionary and trimmed down the list by excluding non-stable-trait, difficult, unfamiliar, redundant, ambiguous, slang, and inappropriate terms. We also included in our initial trait list attributes that are specific for describing faces via literature search. We applied a data-driven approach to select a subset from the initial list to minimize subjective bias in the selection process (**Figure 1b-c**). We showed that our final selected 100 trait attributes were representative of the words English speaking participants used to describe faces spontaneously (**Figure 2a-b**). However, we totally agree with the reviewer that the 100 trait attributes we studied here might not be representative of the terms non-English speakers use to describe trait impressions from faces. We have noted this important limitation and clarified our trait selection method in the revised manuscript (lines 3-9, page 4; lines 4-7, page 21; line 3, page 26).

In response to the reviewer's second concern, we acknowledge that using only Caucasian/white faces is an important limitation of our study, and we agree that we should note this even more explicitly in the paper. This limitation might challenge the generalizability of our findings in two ways. First, faces from other races might possess facial features that are not available in Caucasian/white faces (e.g., the wider noses in the Black population, the flatter faces in the Eastern Asian population), which might produce a bottom-up modification to the psychological space of face judgments. Similarly, our stimuli might have left out many facial features that are unique to other populations that we did not consider, such as unhealthy individuals (e.g., the distinct facial appearance in people with Down syndrome) and very young or very old individuals (e.g., the high and protruding forehead of infants). Second, race as a social construct might also modify the psychological space of face judgments in a top-down manner. We have added emphases on these limitations of our face stimuli in our revised manuscript (line 11, page 4; lines 9-16, page 21).

We also thank the reviewer for pointing out the third limitation in the word embedding computed with deep neural networks. We have noted this limitation in our revised manuscript (lines 7-9, page 21).

In summary, this is a much needed, impressive, highly innovative study with far-reaching results. The paper is well-written. The studies are well designed, and the results are highly relevant to psychologists, neuroscientists, and engineers. I am in strong support for publication.

RESPONSE: Thank you very much.

Some related papers to be considered for additional discussion: Sutherland et al. PNAS

2020, Fan et al., eLife 2020, Srinivasan et al. IEEE TAC 2018, Brooks et al. PNAS 2019, Franklin et al. Emotion 2018.

RESONSE: We thank the reviewer for recommending these highly relevant works. We have cited them in our revised manuscript (line 8, page 3; line 2, page 21; lines 14-20, page 21).

Reviewer #2 (Remarks to the Author):

SUMMARY OF WORK:

Prior work on social facial perception found that a smaller number dimensions underlie the perception (famously, 2 or 3). Similar to this line of work, the authors determined the number of summary dimensions; they selected face images and trait words that are more representative than those used in the prior work, sampling each of them from a more comprehensive set of items evenly from the respective space (face space and trait-word space). Then they determined the number of summery dimensions using multiple methods. In sum, this is a strong piece of work that adds to the field. I only have minor issues with the current versions of the manuscript.

RESPONSE: We thank the reviewer for the positive assessment.

PERSONAL EVALUATION OF WORK:

My initial reaction to the paper was mixed. On one hand, I did not find the results presented not too surprising. Sure, employing a bigger, comprehensive set of words and faces will lead to a larger number of underlying dimensions, explaining a bigger variance in data. It seems to me at first this fact makes this work appear incremental in nature. Relatedly, the selection of 100 words and 100 faces seem arbitrary – why not 200, 400, or 1000 face/words? Following the same logic in the introduction, a larger number of stimuli might have led to a larger number of dimensions (although the authors make a convincing cases of the representative of these stimuli, e.g., Fig 2, Supp Fig 1b).

RESPONSE: The reviewer raises an important point here. We agree with the reviewer that with a more comprehensive set of trait judgments, it is not too surprising to find a larger number of underlying dimensions. However, we find it unlikely that additional dimensions would keep being added if we keep increasing the number of words. Of course, the number of dimensions depends on how much variance one wishes to account for, so if one wants to account for 100% of the variance in the data, the number of dimensions will be the number of words, and so more words will add more dimensions. However, when we decimated our words, we retained our original four dimensions fairly stably, suggesting that indeed these four dimensions are fundamental in some sense.

We also find it very interesting that the actual dimensions themselves (or at least their reasonable interpretation) revealed here are distinct from those proposed in prior work. For example, the warmth dimension we found here and the previously proposed valence dimension each describes different subsets of trait judgments, and the two were only moderately correlated; the competence dimension we found here and the previously proposed dominance dimension each describes different subsets of trait judgments, and the two were not significantly correlated. The discovery of this distinct dimensional space provides a new perspective on how we might understand the relation between the dimensions people use to judge faces and the dimensions people use to judge other people and groups found in previous research outside the field of face perception (e.g., the stereotype content model).

The reviewer raised a good question about the number of selected traits. Ideally, to comprehensively investigate the psychological space of face judgments, one would include as many and various trait words as possible; on the other hand, if different trait words describe very similar trait judgments from faces (e.g., the trait words are very similar semantically), it would be redundant to include them all, and if some trait words are unclear or unfamiliar, obtaining ratings on them would just add noise. Following this logic, we tried to first gather an inclusive trait list from multiple sources, and then applied a data-driven approach to select a subset of traits based on their semantic similarity, clarity, and familiarity. As mentioned, our analyses showed that it is in fact not necessary to include all the 100 trait judgments to yield the four-dimensional space (**Figure 5a** and **Supplementary Table 2b**), and including more words are likely to be redundant and would not alter the dimensions found as also noted by the reviewer (e.g., **Figure 2**).

On the other hand, I find the methodological rigor of the current work impressive and the conclusion fair. (1) The authors carefully set each step of analysis, not leaving anything to guess work. They addressed multiple technical issues that were left not considered in prior work, which might have changed the resulting dimensions and the number of the dimensions, albeit potentially to a small degree. I am talking about careful sampling of representative stimuli, preregistrations, confirmatory analyses, robustness and generalizability testing etc. in the present work. Each of these components add to the current paper's value. (2) The figures are helpful and pleasant to look at. (3) Then there are main analyses that are genuinely novel upon a more careful read, e.g., the calculation of each observer's dimensionality and the assessment of cross-observer reproducibility (Study 2).

When everything is taken into consideration, I believe this is a strong piece of work that adds to the field. I commend the authors.

RESPONSE: Thank you very much for this positive assessment.

MINOR ISSUES:

1. "By analogy, we can perceive (and have words for) many different shades of colors,

but they are all the result of a three-dimensional color space. In the case of color, the answer is easier because we know that there are only three kinds of cones in the retina; in the case of trait judgments of faces, we must infer the psychological space from behavioural data (human subjects' ratings of faces on different trait words)" (p.3)

While this is a compelling analogy, the essential difference between the two systems should be made clear; the cones on the retina have a concrete physical basis, while a n-dimensional trait space does not. The existence of 3 different types of cones explicitly provides the "mechanism" of color perception. On the other hand, n-dimensional trait space (let n be 2, 3, or 4) is simply a statistical summary of behaviour. In other words, the n-d face space is a "description" of a phenomenon (=social facial perception). Based on this, one might even argue that the number of dimensions is unimportant.

RESPONSE: The reviewer raised a very good point. We have clarified the distinction between the two systems in the revised manuscript (lines 15-19, page 3).

2. "true dimensionality" (p.4)

What even is the "true" dimensionality? This is related to my point in MINOR ISSUE No. 1 (the cone and color perception analogy). Personally I am generally uncomfortable with the notion that there are the correct, definite set of underlying dimensions. The dimensions are only a statistical summary of the way we form impressions from faces, influenced by the norm of the time (although according to the paper they seem pretty stable within-observer and cross-culture). I would suggest a rephrase because this like of language ("true") can lead to an essentialist idea and as a result a unfruitful discussion about whether the number of "basic emotion/al expressions" is 4, 5, or 6.

RESPONSE: We thank the reviewer for this cautionary note. We absolutely agree and we have replaced "true" with "comprehensive" (line 1, page 4).

3. "Prior work has reported both common³ and discrepant dimensions in different cultures^{13,14,24,35,37}" (p.15)

The citations should be corrected. For example Ref. no. 13 and 37 only used a English speaking participants in the US and the UK, respectively. As far as I know they did not recruit participants from multiple cultures, and there was no contrast between cultures, which to me seems like a reason you need to remove 13 and 37 from the second list of citations.

RESPONSE: We agree with the reviewer and have removed citation 13 and 37 in the revised manuscript (line 12, page 15).

4. "but, as expected, reproducibility was attenuated by the data quality available (as assessed by within-subject consistency" (p.16) "as well as in individual participants (although this was more difficult to assess, due to data quality" (p.20) and many other places (e.g., p.18)

Here it seems it is assumed that (1) all regions have relatively same level of within-subject consistency and (2) thus a low within-subject consistency reflects data quality of the sample, NOT a genuine evaluation tendency of the sample. While these are reasonable assumptions, there is no way you can confirm these assumptions given the current data – you do not know the true reason of the low within-subject consistency. Thus the mention of data quality seems inappropriate.

RESPONSE: We thank the reviewer for noting this important distinction. We agree that a low within-subject consistency is not equivalent to low data quality, and as pointed out by the reviewer, it is plausible to be a genuine evaluation tendency of the sample. We have removed the claim of data quality throughout the revised manuscript (lines 16-17, page 16; lines 6, 17, 20, page 18; lines 18-19, page 20; line 8, page 34).

REVIEWER COMMENTS

Reviewer #1 (Remarks to the Author):

The authors have addressed all of my comments. I see the authors have also addressed the comments of the second reviewer. Both, my comments and those of the second reviewer, were relatively minor. We both agreed this is highly significant, pressing work. I think the edits made the manuscript much stronger -- especially by citing the limitations of the study so others can pursue follow-ups. All the pieces of this work have been carefully described and evaluated. The results are compelling. No doubt, this will be a very influential paper. The paper is ready for publication.

Aleix M Martinez

Reviewer #2 (Remarks to the Author):

I have no remaining issues. I thank the authors for addressing my previous issues.

Reviewer #3 (Remarks to the Author):

I advise 'revise and resubmit'. The article sheds new light on a fascinating and potentially important research question: How do humans attribute traits to another person based on simply looking at their face? The study innovates on previous research by increasing the scale of the measures and introducing new methods with machine learning. However, there are some issues that need to be addressed before it would be ready for publication.

Most importantly, the authors need to clarify the status of the study with respect to two methodological limitations: the reliance on English and the reliance on 'white' faces. Second, they need to clarify more clearly what the relevance and implications of the findings are for potential applications.

Below find some specific comments on the paper that elaborate on the issues that I suggest should be addressed. (Note: My comments do not address matters relating to the statistical or computational methods as they are not in my area of expertise.)

1.

The article claims to have discovered 'four dimensions characterizing comprehensive trait judgments of faces', as reflected in the article title. I see two issues with this claim, both of which stem from the implication that the findings are of universal relevance; i.e., that the discoveries—(a) about faces and (b) about trait judgements—are about humans rather than some subset of humans. The experiment correlates words with faces. The language of the article implies universal relevance of the findings: eg the abstract opens, 'People readily attribute many traits to faces'. The generic use of 'people' and 'faces' suggests 'all/any people' and 'all/any faces'. This immediately raises the question: How can conclusions about human behavior in general be derived from research limited to inputs and measures that are representative of only a small subset of humanity?

2.

The experiment's sample of words is not representative of human languages in general.

What reason is there to think that English provides appropriate categories for pan-human distinctions? English is just one of the 6000 or so languages spoken on Earth today. There is some acknowledgement of this in the manuscript but it is too buried. Line 324: 'We note that our sampling of trait words was limited to English words'. Why the limitation to English? Languages have thousands of adjectives and there are thousands of languages in the world. Why is English regarded as an appropriate measure for a pan-human propensity? Or is this intended to be a study of English? If so, this should be made clear. A straightforward solution would be to state that this is a study of English-language attribution (eg by putting 'English' in the title). But this raises another problem: Spanish was used in the Peru data. If the researchers considered it tolerable to use translations of English into other languages, why wasn't this done with many more languages? EG Why no translation to Hindi in India etc. (However, I emphasize that it is well established that languages do not have many translational equivalents in their vocabularies at all, let alone in their sets of adjectives labeling personality traits.)

3.

Again on the matter of language, and the reliance on English, there is a real concern about the validity of English as a measure across the populations tested. The judgments in Latvia, Peru, the Philippines, India, Gaza, were done in English by non-native speakers. How comparable are these judgments across the various countries? How comparable are they to the native English speaker judgments? Methods section (line 565) says subjects had to be 'proficient in English' (up to high school level). Exactly how 'proficient' did somebody have to be? Even with someone who is truly 'proficient', can we be confident that they understand the 100 words, even the four main trait words, in the same ways that English native speakers do? An additional criterion was that they must 'have never visited or lived in Western-culture countries', line 567); this almost guarantees that their English proficiency would be far from native-speaker level.

4.

The Peru participants had the words translated into Spanish. There is no question that few if any of the words in the list have direct translational equivalents in Spanish. How do the authors respond to this?

5.

The experiment's sample of faces is not representative of human faces in general; they were all 'white, neutral, adult' faces. Line 399: 'Our goal was to derive a representative set of neutral, frontal, white faces'. Lines 406-7: 'We further restricted ourselves to images of Caucasian adults and neutral expression.' What is the reason for this restriction to 'white' faces? The choice exposes the paper to the obvious critique that the study assumes 'white' people to be representative of humans more generally. The authors need to give good reasons why 'white'-only faces were deemed appropriate for this study, or they should explicitly frame the study as being a study of judgements about 'white' faces, not about human faces. To be clear, either you are saying that 'white' faces are representative of human faces, and so we are justified in making claims from these data about people's judgements of 'faces' in general, or the study is about 'white' faces (not 'faces') and this should be studiously noted in the paper title and through the paper, in order to avoid misunderstanding and misrepresentation of the study. (But then this raises the serious question: Why conduct a study of 'white' faces in particular?)

Also note that the status of a 'white' face will be different for respondents in the US as opposed to Peru, the Philippines, and India, for the simple reason that 'white' faces are

'marked' (e.g., because they are less frequent) for members of those populations in a way that they are not 'marked', say, for US populations. This raises another issue of comparability in the data.

6.

It must be made crystal clear that this study is about people's JUDGMENTS OF faces, and not a claim about the true qualities of individuals with certain faces. The study's findings concern human judgments of the traits of people with certain faces, and they do NOT concern the true traits of people with certain faces. Of course this is clear to the authors but it must be emphasized at the highest level, because it can be almost guaranteed that media and social media attention to this article will misconstrue this point. While this paper makes claims about people's evaluations of faces, it will be erroneously described as making claims about a link between faces and personality traits, which in turn would imply, for example, that AI applications should be able to use facial recognition to scan faces in a group and determine who is 'artistic', who is 'innovative', who is 'competent', who is 'warm', etc. I am not saying that this is not a possible claim, nor that it should not be made if there were evidence for it, but that the authors should be very clear what they are and are not saying. I acknowledge that the authors are consistent in referring to 'trait judgments of faces' rather than 'traits of faces', but still I would recommend flagging this issue explicitly at the beginning of the abstract and certainly in any media release with a statement along the following lines: 'It is impossible to discern someone's personality traits from their face, yet people nevertheless confidently make trait judgements of people based on their face alone. Here, we want to understand the principles behind those judgments...'

To give the authors a sense of what they are likely to confront, it would not be surprising to see a reaction such as the following (though I emphasize that this study clearly does not claim to be able to predict traits from faces, rather I am pointing to likely interpretations from lazy readers):

<https://www.inputmag.com/culture/researchers-still-foolishly-think-ai-can-predict-criminality-by-looking-at-photos>

Paper here: <http://archive.is/N1HVe#selection-1609.160-1609.316>

An example that is closer to this article is in the recent paper in Nature Communications on faces and trustworthiness in historical paintings:

Safra, L., Chevallier, C., Grèzes, J. et al. Tracking historical changes in trustworthiness using machine learning analyses of facial cues in paintings. Nat Commun 11, 4728 (2020). DOI: 10.1038/s41467-020-18566-7

A social media response claimed that this study was 'racist' and 'phrenology-adjacent':

<https://medium.com/@rory.spanton/a-top-scientific-journal-just-published-a-racist-algorithm-b13c51c4e5b0>

<https://www.inputmag.com/culture/this-algorithmic-study-about-trustworthiness-has-some-glaring-flaws>

As the mistaken responses to the Safra et al 2020 paper reveal, the authors of this article will need to take extraordinary care to show that they are investigating people's perception of

traits based on a face and not the actual traits of a person with that face. (I note that this can readily be famed as an important step in addressing problems of discrimination based on appearance.) The mismatch between these two things, and the danger of confusing them, should be foregrounded in the discussion not only in order to avoid the accusation of doing 'phrenology-adjacent' or 'racist AI' research but also to point to an important value of this study. In other words, don't avoid the discussion of human difference and pernicious bias in judgement, instead embrace those things and articulate how the study can help. The fact that people judge others by their faces arguably contributes to enormous social problems and lack of fairness. These findings could be framed as taking us a step closer to understanding how to better understand the problem, and, potentially, remedy it. It could also be framed as a positive feature that the faces are 'white' because this is an attempt to control the study for factors of racial discrimination.

7.

I note that the faces themselves do not all look obviously 'white'—I counted at least eight that could fit the description 'person of color' of one kind or another.

8.

Clarify that the four factors correspond to the actual words used in judgments, they are not the authors' glosses of four abstract factors drawn from the model. I note with interest that the four factors all seem positively-valenced. Will the authors explain or speculate why these four traits seem important as discriminating factors? And are they implying that we only need these four words in order to most efficiently discriminate among face-trait judgements? What does it imply about how people distinguish or judge faces in real life?

9.

Note that in evolutionary terms, this study tests an unusual task, because in the ancestral environment we would not have been confronted by unfamiliar faces very often; would the authors argue that these judgement tendencies would be adaptive in the ancestral human context?

10.

The discussion falls flat, by merely gesturing toward future work that would address shortcomings of this study. But there are interesting implications and they should be addressed. For example: what is the relation between people's trait judgements and the real traits of people? There is a mismatch in social discourse around human traits: on the one hand, people appear to consistently judge certain faces as appearing 'competent', but on the other hand people want to say that it is impossible to tell from somebody's face whether they are actually a competent person. How can this research help counteract the human propensity to judge people by their faces? The study should finish with some straightforward questions to be asked next, not a request that subsequent studies should address shortcomings of this one.

11.

The discussion should address explanations: WHY would people have strong and consistent judgments about whether a stranger is 'abusive' or 'conscientious' based on their face? Are these judgments hardwired? Are the judgments quickly overwritten when we get to know the person? Or is it just that we are wired with a drive to classify faces in terms of traits and the actual mapping of faces to traits is a result of historical contingencies, eg, tropes based on movies (look at Bond villains, horror films, etc. over the decades)? Are the mappings learned? These are crucial questions and the study doesn't address them.

12.

Line 326: 'additional research will be needed to elucidate how language might modify the psychological space of face judgments'. A relevant reference to note is the finding that describing a face in words causes the speaker to be worse at remembering and later identifying the face (by a process of 'verbal overshadowing'):

Schooler, Jonathan W, and Tonya Y Engstler-Schooler. 1990. "Verbal Overshadowing of Visual Memories: Some Things Are Better Left Unsaid." *Cognitive Psychology* 22 (1): 36–71. [https://doi.org/10.1016/0010-0285\(90\)90003-M](https://doi.org/10.1016/0010-0285(90)90003-M).

Alogna, V. K., M. K. Attaya, P. Aucoin, Š. Bahník, S. Birch, A. R. Birt, B. H. Bornstein, et al. 2014. "Registered Replication Report: Schooler and Engstler-Schooler (1990)." *Perspectives on Psychological Science* 9 (5): 556–78. <https://doi.org/10.1177/1745691614545653>.

Reviewer #4 (Remarks to the Author):

The authors have an ambitious and highly interesting research aim which uses word reduction techniques to generate a small set of non-redundant trait words, approaching the issue of impression formation from a different angle than previous work, which has focused on spontaneously attributed impressions of faces. In many ways, the work is reminiscent of a modern take on Allport's original, classic studies into trait impressions in the 30s (Allport & Odbert, 1936), albeit with up-to-date statistical analyses and tools which were not available in Allport's day. To be clear, I am in favour of publication and think the aims and methodological rigour of the work alone makes it worth publication. My comments are largely aimed at addressing the theoretical contribution, and in particular I would like the authors to acknowledge existing theory in more depth.

First, are the dimensions of warmth and competence really that different from valence and dominance? I wasn't convinced that the differences were as large as stated and the authors risk overextending their interpretation here. The Stereotype Content Model also shares theoretical similarities with Alex Todorov's model which go beyond the specific labels of warmth etc; i.e. both suggest that at the core, trait judgements are based on approach/avoid orientation. Particularly, I would urge the authors to consult previous studies which have also investigated the question how warmth and competence relate to trustworthiness and dominance and/or across populations of faces; Walker et al 2017, JPSP, Sutherland et al 2016, Cognition, 2018, PSPB Collova et al 2019, JPSP. Some of these studies find greater similarity than others, and particularly, dominance and competence appear to diverge more for populations other than White males (i.e. dominance appears to be less important for characterising female faces, Asian faces, and children's faces). The authors did a good job of acknowledging the limitations of their face stimuli in the abstract/discussion, but I would like to see greater discussion of this point which includes this previous literature. The authors could also consider the question of whether trait space is actually more flexible than previously recognised (Stolier et al, 2018, TICS).

Similarly, it looked like the "youth" factor in the 4D solution appeared to replicate the "youthful-attractiveness" factor found by Sutherland et al. (2013), but the authors could clarify here. Presumably there is a strong conceptual similarity if both rely on age, but where does attractiveness sit in the four dimensions? Attractiveness looked to cross-load in the

three dimensional replication, perhaps an indication that age is the more robust or distinctive aspect of this dimension? (also perhaps why this third dimension appeared in our 2013 study with more variable ambient images that differed on age, but not in Alex Todorov's original work, where age was not varied).

Second, I am very reassured that the authors replicate their work with multiple modelling methods, including EFA with oblimin and PCA, as well as a new nonlinear technique. This paper will have an important contribution to make given the current debate over the best approach for dimensionality reduction across culture (e.g. Jones et al Nature Human Behaviour, in press, Oh and Todorov, in press). We have followed a similar logic to assessing whether models are robust and similarly did not find large differences between analyses. I also appreciated the replication with reference to previous published models, as this approach was very helpful in outlining similarities and differences across studies.

Finally, it is also reassuring that the authors broadly replicated their models across culture. The cross-cultural work speaks to a current question in the literature as to how universal these trait impressions are (Jones et al Nature Human Behaviour, in press, Walker et al 2011, SPSS, Sutherland et al 2018, PSPB, Zebrowitz et al J Cross Cultural Psychology, 2012). The work of Mirella Walker and her colleagues should again be noted here; her team was arguably the first to test face trait models across culture. Leslie Zebrowitz also carried out outstanding work into impressions from faces made by non-Western cultures, a research effort which I would also like the field to acknowledge more clearly when culture is investigated.

Clare Sutherland

Minor issues: The authors are to be commended for investigating this question at the individual participant level, as it involves many hours of testing to get a full set of data from one participant; moreover, they managed to get samples across culture here too. We (Sutherland et al 2020, BJP) also managed to reproduce a particular (3D) model at the individual level in a handful of participants with a smaller set of traits, but the current paper goes far beyond what we attempted. Really nice work.

Could the focus on sampling words (as is typically carried out in social cognition research), rather than faces (as is typically carried out in face perception studies), have led to potential differences across trait and face models i.e. warmth v trust? I think not, given the supplementary analyses, but the authors may wish to specify, as this question was previously unclear e.g. Sutherland et al 2016, Cognition.

Carryover effects (Rhodes et al 2006). Participants rated multiple traits, so I did wonder if this procedure could affect the results. Is it possible to examine the trait space with just the first rating given by participants? Does it change anything?

Line 622 - Hehman et al (2017, JPSP) also showed that traits relating to physical aspects showed higher between-subjects consistency than more abstract traits.

I thought it was a service to the field that the authors recommend a subset of 18 words which could be used as a shorthand, but it would also be useful to know which words absolutely had the best measurement invariance (i.e. what if studies can only recruit ten words, or even only four words?)

I'm not sure I followed Fig 7. Which dimensions are which in this figure? Is dimension 3, "femininity", and dimension 4, "youth"? Or are they different across cultures?

Line 103 ICCs (and in the methods) - which ICCs specifically? The authors could give more justification and relate to a specific case model from McGraw and Wong (1996). I would prefer a two-way random model - subjects in columns should also be random for example (see p37). i.e. a two-way random model (C, 1), is the closest approximation to alpha at the individual level. See also p38. The authors may indeed have used this ICC so it may just be a case of clarification.

REVIEWER COMMENTS

Reviewer #1 (Remarks to the Author):

The authors have addressed all of my comments. I see the authors have also addressed the comments of the second reviewer. Both, my comments and those of the second reviewer, were relatively minor. We both agreed this is highly significant, pressing work. I think the edits made the manuscript much stronger -- especially by citing the limitations of the study so others can pursue follow-ups. All the pieces of this work have been carefully described and evaluated. The results are compelling. No doubt, this will be a very influential paper. The paper is ready for publication.

Aleix M Martinez

RESPONSE: We thank the reviewer for the positive assessment.

Reviewer #2 (Remarks to the Author):

I have no remaining issues. I thank the authors for addressing my previous issues.

RESPONSE: We thank the reviewer for the positive assessment.

Reviewer #3 (Remarks to the Author):

I advise 'revise and resubmit'. The article sheds new light on a fascinating and potentially important research question: How do humans attribute traits to another person based on simply looking at their face? The study innovates on previous research by increasing the scale of the measures and introducing new methods with machine learning. However, there are some issues that need to be addressed before it would be ready for publication.

Most importantly, the authors need to clarify the status of the study with respect to two methodological limitations: the reliance on English and the reliance on 'white' faces. Second, they need to clarify more clearly what the relevance and implications of the findings are for potential applications.

RESPONSE: We thank the reviewer for the expert critique. We have now emphasized our limitations on the reliance on English and white faces in the Abstract, Introduction, and Discussion. We have also expanded our statements of other limitations and clarified the implications of our findings in the revised Discussion, and in the new Impact Statement requested by the editorial team. We respond to the reviewer's comments raised below in detail.

Below find some specific comments on the paper that elaborate on the issues that I

suggest should be addressed. (Note: My comments do not address matters relating to the statistical or computational methods as they are not in my area of expertise.)

1.

The article claims to have discovered ‘four dimensions characterizing comprehensive trait judgments of faces’, as reflected in the article title. I see two issues with this claim, both of which stem from the implication that the findings are of universal relevance; i.e., that the discoveries—(a) about faces and (b) about trait judgements—are about humans rather than some subset of humans. The experiment correlates words with faces. The language of the article implies universal relevance of the findings: eg the abstract opens, ‘People readily attribute many traits to faces’. The generic use of ‘people’ and ‘faces’ suggests ‘all/any people’ and ‘all/any faces’. This immediately raises the question: How can conclusions about human behavior in general be derived from research limited to inputs and measures that are representative of only a small subset of humanity?

RESPONSE: The reviewer’s points are very well-taken. We previously acknowledged these limitations in the last sentence of our Abstract, but we fully agree with the reviewer that more is needed. We have made changes to Title, Abstract, Introduction, Discussion, and added a new “Impact Statement” in the supplementary materials. Briefly, here is a summary of the changes we have made.

Title: We feel that the title should be as simple as possible without giving any wrong ideas – in particular, we have removed the word “comprehensive” from the title, and used “attributions” to emphasize that the study is about people’s opinions about others’ traits instead of others’ actual traits, but otherwise continue to leave the other parts without explicitly listing limitations of the study in the title.

Impact statement: As requested by the editor, we now include a lengthy “impact statement” where we elaborate on the limitations of our study, which are in fact even more severe than what the reviewer suggests. In brief, it is theoretically and practically challenging to define the domain of representative face stimuli: we do not know what faces our participants see in everyday life, even less what faces they saw when growing up, and even less what faces their ancestors might have seen that led to the evolution of the psychological dimensions we describe (related to the point raised by the reviewer below). Similar challenges are posed for the words and concepts that describe traits. Like the reviewer, we feel that a full framing our findings will be an important reminder for the field, and the new “impact statement” now gives us the space to elaborate on this.

Abstract: We have revised our Abstract substantially. In particular, we added a second sentence that immediately modifies the introduction of the first sentence, noting the limitations the reviewer raises, “...*the details of attributions depend on the language available to describe social traits, and on the faces...*”. The first sentence itself, the statement that “people readily attribute traits to faces,” we take to be uncontentious because it does not make any claims about specific people, traits, or kinds of faces – in

the absence of evidence that there are cultures that fail to make social attributions from faces, we take this to be a reasonable opening to introduce the topic of our paper. We now have included a phrase stating that those attributions are often inaccurate in that opening sentence. We have also clarified that only English trait words and white faces were sampled in our study. We note that the Abstract's 150-word limit puts constraints on how comprehensively we can list the limitations of our study, regarding which we provided more detailed treatment of this topic in the revised Introduction and Discussion.

Introduction: we have added a number of cautionary statements to the Introduction. The first paragraph notes that, "*Although trait attributions from faces may not reflect people's actual traits and reveal more about our own biases and stereotypes*^{3,6,7...}" (page 3, lines 7-8); the third paragraph mentions repeatedly that we used English words and white faces, and provides brief explanation on such limitations, "*We focus on English words because English is the most-spoken language (native and learned) around the world, which makes replicating the study in other parts of the world more feasible (Study 2). We limit ourselves to frontal, neutral faces of white, working-age individuals in an attempt to control for factors such as racial and age discrimination, which are known to bias face perception*^{23,40-43}. Relatedly, this restriction of the variance in our face stimuli served to increase statistical power, by eliminating factors that our study did not intend to investigate, such as facial expressions (see Methods)." (page 4, lines 11-17), which we elaborated more in the revised Discussion.

Discussion: we have further elaborated the limitations of our study in the revised Discussion, citing additional work from linguistics. For instance, we have added two paragraphs that emphasize that no implications about people's actual traits should be drawn from our study, which is about people's opinions about others' traits (page 21, lines 16-23; page 22, lines 1-7), and that our study is "*biased with respect to the participants, the trait words, and the faces... our study does not, and cannot, make any claims about Human Nature or universality*" (page 22, lines 8-21). We further expanded on the effects of language, citing work from linguistics recommended by the reviewer (page 22, lines 22-23; page 23, lines 1-10), and on the various factors that might shape face perception (page 23, lines 11-21).

Throughout the paper, we have gone through everywhere and tried to remove any misleading statements that might suggest generalizability inappropriately. For instance, the entire section for Study 2, which used to have the subheading "Generalizability across countries and regions", now has the more neutral subheading, "Results from other countries" (page 15, line 16), and we have tried throughout not to give any false suggestions of generalizability or universality.

2.

The experiment's sample of words is not representative of human languages in general. What reason is there to think that English provides appropriate categories for pan-human distinctions? English is just one of the 6000 or so languages spoken on Earth today. There is some acknowledgement of this in the manuscript but it is too buried.

Line 324: 'We note that our sampling of trait words was limited to English words'. Why the limitation to English? Languages have thousands of adjectives and there are thousands of languages in the world. Why is English regarded as an appropriate measure for a pan-human propensity? Or is this intended to be a study of English? If so, this should be made clear. A straightforward solution would be to state that this is a study of English-language attribution (eg by putting 'English' in the title). But this raises another problem: Spanish was used in the Peru data. If the researchers considered it tolerable to use translations of English into other languages, why wasn't this done with many more languages? EG Why no translation to Hindi in India etc. (However, I emphasize that it is well established that languages do not have many translational equivalents in their vocabularies at all, let alone in their sets of adjectives labeling personality traits.)

RESPONSE: We agree with the reviewer that our study's sample of words is not representative of human languages in general. The reason for limiting ourselves to English is twofold: (1) that English is the most commonly spoken language (including native and learned) around different parts of the world, making it feasible to replicate our study using the exact same experimental procedure in as many different samples around the world as possible (Study 2); (2) the comprehensive list of attribute words from which sampled was available only in English-based psychology studies, and similarly the machine learning approaches for sampling the words depended on deep neural networks that were trained on the corpus of English words. As well, the common language of the investigators and our colleagues was English and it was simply not our intent to conduct a study of other languages than English. We have now clarified this in the revised Introduction (page 4, lines 11-13), and expanded more on our limitations regarding language in the Discussion (page 22, lines 22-23; page 23, lines 1-10).

Our original intent of recruiting different samples around the world, as we clarify in the paper (page 16, lines 1-3), was not to provide a cross-cultural or cross-linguistic study in our Study 2, but mere to help expand the reliability of our findings by reproducing them in samples additional to American subjects tested over the internet (our Study 1), which has been the modal approach taken by all studies in this field. To that end, we tested subjects in 7 different countries in person in very dense sampling. Thus, Study 2 was not and is not intended to be a cross-cultural study, and we don't make any claims about cultural or linguistic universality, and instead note the limitations (Abstract; page 3, lines 13-14; page 4, lines 11-13; page 22, lines 10-13; page 23, lines 1-10).

In Study 2, we intended to recruit primarily participants who speak English precisely for the reasons noted by the Reviewer: so that we could avoid the issue of translation. We had to switch to Spanish for participants in Peru because we were not able to recruit the target sample size of participants who speak English there, and we felt that collecting and presenting the data from Spanish-speaking participants would still be useful to include in our paper.

To alleviate the issues with translational equivalents and potential different interpretations of the trait words by participants who speak different (native) languages

(and even within participants who are native English speakers), we did not simply give subjects a single trait word in isolation. Instead, for each trait word, we provided a brief definition of what the word means in the context of trait attributions from faces (Supplementary Table 1). This is also an improvement over much prior work and important to note: participants were not merely given words to rate, but definitions of their meaning.

Importantly, our study is not merely about the trait words we used. The words are one way of getting at the psychological concepts people use to make attributions of faces. In principle, one could imagine a study such as ours, with the same conclusions, that is based not on using words at all (e.g., using pairwise similarity judgments about the faces or other approaches). We tested the possibility that the psychological dimensions we found based on the trait ratings from faces are equivalent to or constrained by the semantic relations between the words. That turned out not to be the case: dimensionality reduction based merely on the semantic similarity among the words (without any attributions from the faces) suggests a much higher number and a different set of dimensions than the four dimensions we found with faces (see Methods, subsection “Sampling of trait words”, page 26, lines 21-22 and page 27, lines 1-13).

Relatedly, our findings showed that, even with the limitations on language, the same four psychological dimensions were found in samples that speak different native languages (e.g., the Latvia sample) and the Peru sample that used Spanish to complete the experiments, suggesting that there is similarity in the psychological dimensions people across different regions use to represent the faces. As noted above, a stronger test of such a claim might be obtained in future studies using experiment paradigms that do not involve words at all (e.g., have participants rate the pairwise similarity of all faces or sort the faces into piles based on what type of person they look like). We have noted this in the revised Discussion, *“These findings suggest that while languages likely shape the words and concepts available to describe individual trait attribution, the psychological dimensions (which capture the relationship between groups of trait attributions) that people use to represent face attributes are to some extent similar across different countries. This could be tested further in future studies that use no words at all, and instead use measures such as pairwise similarity judgments or pile sorting of the faces⁶⁸.”* (page 21, lines 1-5).

3.

Again on the matter of language, and the reliance on English, there is a real concern about the validity of English as a measure across the populations tested. The judgments in Latvia, Peru, the Philippines, India, Gaza, were done in English by non-native speakers. How comparable are these judgments across the various countries? How comparable are they to the native English speaker judgments? Methods section (line 565) says subjects had to be ‘proficient in English’ (up to high school level). Exactly how ‘proficient’ did somebody have to be? Even with someone who is truly ‘proficient’, can we be confident that they understand the 100 words, even the four main trait words, in the same ways that English native speakers do? An additional criterion was that they must ‘have never visited or lived in Western-culture countries’, line 567); this almost

guarantees that their English proficiency would be far from native-speaker level.

RESPONSE: We understand the reviewer's concern. We did not measure exactly how proficient in English the participants were. As mentioned above, we did provide the definition of each of the words to help participants understand what the words mean. We acknowledge that even providing the definitions still would not have ensured that our participants in Study 2 understood each of the words in exactly the same way as the participants in Study 1 did. For that matter, there would probably be some heterogeneity even in how native English speakers in Study 1 understood the words. We acknowledge that there are additional sources of variance, depending on each participant's vocabulary and conceptual sophistication.

If language had been a problem, both in terms of participants who are non-native English speakers not understanding the meaning of our trait words (and the one sentence definitions that we provided as mentioned above) and in terms of participants who speak different languages understanding the meaning of our trait words (and the one sentence definitions that we provided) differently, this would have made it more difficult for us to reproduce the four dimensions found in Study 1.

Nonetheless, the findings bear out the fact that, first, we found that the trait ratings collected from these 7 samples in Study 2 had satisfactory within-subject consistency and between-subject consensus (page 37, lines 10-23; page 38, lines 1-6). This confirms that the attributions our participants made were measuring something meaningful and reliable about both the faces and the words. Second, across all these different subject samples, notwithstanding possible differences in English proficiency, we found high representational similarity (page 16, lines 6-12) and recover the same four factors (Fig. 6), suggesting that even though people speaking different languages may have different words that can describe any single trait, they still share something in common in their psychological representation of the faces. As the analogy we gave in the Introduction notes (page 3, lines 15-20), people speaking different language might have different words and concepts for different shades of colors (indeed, it is known that they do), but they are all the result of a three-dimensional color space. That being said, we do acknowledge that the dimensions of trait attributions from faces are less straightforward than the dimensions of colors (since the former has to be inferred from human subject's ratings), and language would likely shape the space (page 23, lines 1-10).

To reiterate, Study 2 was not and is not intended to be a cross-cultural study. If we made strong claims about cultural differences and non-generalizability in our paper, the reviewer's points would indeed be confounds, but we have clarified in the paper that we are not making any such claims (page 21, lines 12-15). Study 2 is intended to help expand the reliability of our findings by reproducing them in samples additional to the American subjects tested over the internet (Study 1).

4.

The Peru participants had the words translated into Spanish. There is no question that

few if any of the words in the list have direct translational equivalents in Spanish. How do the authors respond to this?

RESPONSE: We agree with the reviewer that probably few of the English trait words have direct translational equivalents in Spanish. We paid an effort to alleviate this issue by providing not only the single trait words, but also a definitional sentence that describe what each word means (Supplementary Table 1), which should make it more feasible to communicate the meaning of a word across different languages.

Furthermore, the data bear out the fact that, notwithstanding possible differences in language, the correlation structure of attributions from faces across trait words was highly similar between the Peru sample and the Study 1 sample ($r = 0.85$). In addition, as mentioned above, we showed that the four psychological dimensions are not simply semantic dimensions of the trait words (page 26, lines 21-22; page 27, lines 1-13); they carry information about people's mental representation of the faces, and the same four dimensions were found in the Peru sample (Supplementary Fig.6c and Fig.6b).

Finally, we reiterate that we do not make any claims about the equivalence of the words, or derive conclusions about cultural differences that could be confounded by such differences.

5.

The experiment's sample of faces is not representative of human faces in general; they were all 'white, neutral, adult' faces. Line 399: 'Our goal was to derive a representative set of neutral, frontal, white faces'. Lines 406-7: 'We further restricted ourselves to images of Caucasian adults and neutral expression.' What is the reason for this restriction to 'white' faces? The choice exposes the paper to the obvious critique that the study assumes 'white' people to be representative of humans more generally. The authors need to give good reasons why 'white'-only faces were deemed appropriate for this study, or they should explicitly frame the study as being a study of judgements about 'white' faces, not about human faces. To be clear, either you are saying that 'white' faces are representative of human faces, and so we are justified in making claims from these data about people's judgements of 'faces' in general, or the study is about 'white' faces (not 'faces') and this should be studiously noted in the paper title and through the paper, in order to avoid misunderstanding and misrepresentation of the study. (But then this raises the serious question: Why conduct a study of 'white' faces in particular?)

Also note that the status of a 'white' face will be different for respondents in the US as opposed to Peru, the Philippines, and India, for the simple reason that 'white' faces are 'marked' (e.g., because they are less frequent) for members of those populations in a way that they are not 'marked', say, for US populations. This raises another issue of comparability in the data.

RESPONSE: This is an excellent point. First, we have gone through the paper and tried to remove any misleading statements that might suggest universality or generalizability

inappropriately. For example, in the Title, we removed the word “comprehensive”. In the Introduction, we emphasized that our face images were sampled to be representative of the physical structure of white, adult faces (Fig.2c-d; Supplementary Fig1.b), not to be representative of all human faces. In the Results, we have refrained from using the word “generalizability” and claiming that Study 2 suggests generalizability; instead, we now describe these results as support for the reliability of our findings which showed that the four-dimensional space could be reproduced in additional samples across different regions besides Study1. In the Discussion, we have expanded on discussing the limitations of our study, including the ones regarding the faces, and have stated explicitly that “*our study does not, and cannot, make any claims about Human Nature or universality*” (page 22, lines 8-21), and that the generality of our findings are constrained (page 23, lines 11-21).

Second, we have now provided an explanation for using white faces in the Introduction (page 4, lines 13-17), and elaborated more on the limitations of our face stimuli in the Methods (page 27, lines 15-20) and Discussion (page 24, lines 11-21). In particular, the reasons for constraining ourselves to white faces are two-fold: (1) Scientific—racial bias and stereotypes in face perception have been demonstrated by a large literature (as noted below by the reviewer); we expect that other-race faces, and similarly faces of children or infants, faces of sick or injured individuals, etc. would all receive different ratings because of these additional factors that would be evaluated. We wished to avoid, e.g., other-race effects, in good part in order to control for those additional source of variance, and achieve the statistical power we needed for dimensionality reduction; (2) Practical—we need a sufficient number of high-quality face images for representative sampling, and faces of white individuals were most available in extant databases at the time the study was conducted. For instance, in the Methods, we note that “*We aimed to maximize variability in facial structure (distinct looking individual faces) while controlling for factors such as race, expression, viewing angle, gaze, and background, which our present project did not intend to investigate and which would reduce statistical power due to additional degrees of freedom.*” (page 27, lines 15-20).

Third, as noted by the reviewer, the status of different faces will be different for people in different regions due to the discrepancy in what faces people are frequently exposed to. Indeed, it is a challenge to define the domain of faces that are representative of all human faces: should they be the faces that were seen by our ancestors, experienced during development, or seen during the lifetime of specific participants? We have now expanded on this important issue in our Discussion (page 22, lines 13-22) and “impact statement”.

Finally, as stated above, we do not intend to compare results across cultures in Study 2, and we don’t make any claims about cultural differences or universality. Study 2 is intended to help expand the reliability of our findings by reproducing them with additional samples that are in as different regions around the world as possible.

6.

It must be made crystal clear that this study is about people’s JUDGMENTS OF faces,

and not a claim about the true qualities of individuals with certain faces. The study's findings concern human judgments of the traits of people with certain faces, and they do NOT concern the true traits of people with certain faces. Of course this is clear to the authors but it must be emphasized at the highest level, because it can be almost guaranteed that media and social media attention to this article will misconstrue this point. While this paper makes claims about people's evaluations of faces, it will be erroneously described as making claims about a link between faces and personality traits, which in turn would imply, for example, that AI applications should be able to use facial recognition to scan faces in a group and determine who is 'artistic', who is 'innovative', who is 'competent', who is 'warm', etc. I am not saying that this is not a possible claim, nor that it should not be made if there were evidence for it, but that the authors should be very clear what they are and are not saying. I acknowledge that the authors are consistent in referring to 'trait judgments of faces' rather than 'traits of faces', but still I would recommend flagging this issue explicitly at the beginning of the abstract and certainly in any media release with a statement along the following lines: 'It is impossible to discern someone's personality traits from their face, yet people nevertheless confidently make trait judgements of people based on their face alone. Here, we want to understand the principles behind those judgments...'

To give the authors a sense of what they are likely to confront, it would not be surprising to see a reaction such as the following (though I emphasize that this study clearly does not claim to be able to predict traits from faces, rather I am pointing to likely interpretations from lazy readers):

<https://www.inputmag.com/culture/researchers-still-foolishly-think-ai-can-predict-criminality-by-looking-at-photos>

Paper here: <http://archive.is/N1HVe#selection-1609.160-1609.316>

An example that is closer to this article is in the recent paper in Nature Communications on faces and trustworthiness in historical paintings:

Safra, L., Chevallier, C., Grèzes, J. et al. Tracking historical changes in trustworthiness using machine learning analyses of facial cues in paintings. Nat Commun 11, 4728 (2020). DOI: 10.1038/s41467-020-18566-7

A social media response claimed that this study was 'racist' and 'phrenology-adjacent':

<https://medium.com/@rory.spanton/a-top-scientific-journal-just-published-a-racist-algorithm-b13c51c4e5b0>

<https://www.inputmag.com/culture/this-algorithmic-study-about-trustworthiness-has-some-glaring-flaws>

As the mistaken responses to the Safra et al 2020 paper reveal, the authors of this article will need to take extraordinary care to show that they are investigating people's

perception of traits based on a face and not the actual traits of a person with that face. (I note that this can readily be framed as an important step in addressing problems of discrimination based on appearance.) The mismatch between these two things, and the danger of confusing them, should be foregrounded in the discussion not only in order to avoid the accusation of doing ‘phrenology-adjacent’ or ‘racist AI’ research but also to point to an important value of this study. In other words, don’t avoid the discussion of human difference and pernicious bias in judgement, instead embrace those things and articulate how the study can help. The fact that people judge others by their faces arguably contributes to enormous social problems and lack of fairness. These findings could be framed as taking us a step closer to understanding how to better understand the problem, and, potentially, remedy it. It could also be framed as a positive feature that the faces are ‘white’ because this is an attempt to control the study for factors of racial discrimination.

RESPONSE: We are grateful for these important comments, and fully agree with the reviewer. We’ve highlighted that the study is about making attributions from faces and that these attributions are likely invalid as concerns people’s actual traits right at the beginning of the Abstract, “*People readily (but often inaccurately) attribute traits to others based on faces...*”, and in the first paragraph of the Introduction, “*Although trait attributions from faces may not reflect people’s actual traits and reveal more about our own biases and stereotypes...*” (page 3, lines 7-8).

We further clarify the mismatch between the two in the revised Discussion. For example, in a revised section (page 21, lines 16-23; page 22, lines 1-7), we state, “*There are several important limitations of our findings, some of which we elaborate on in the Impact Statement in the Supplementary Materials. First, it is critically important to reiterate that our study is one of people’s attributions or opinions about faces, which are generally thought to reflect substantial biases and not be valid^{3,71}. That is, whether somebody’s face looks competent has little to do with whether or not they are actually competent. In fact, it remains challenging how even to define a “ground truth” here since many social attributions are fundamentally subject to particular occasions and opinions.*” We fully agree with the reviewer that trait attributions from faces like the psychological dimensions we investigated here are a reflection of bias and stereotyping; we also highlight this in the Methods, section “Labeling of Dimensions” (page 33, lines 7-15). We will also pay attention to this issue in any media release as the reviewer advised.

We thank the reviewer for noting that one reason for choosing white faces is to eliminate a source of unwanted variance – namely race bias. Indeed, that is one main reason we chose to use white faces, and we now clarify this in the revised Introduction (page 4, lines 13-17).

We also appreciate the reviewer’s input on the potential positive implications of our findings—as a step closer to understanding these stereotypes and to figure out the remedy. We have now incorporated this in our Impact Statement, “*Indeed, our findings identify four important dimensions that may contribute to biases and stereotypes that people exhibit when viewing faces, informing future work on stereotyping.*”

7.

I note that the faces themselves do not all look obviously ‘white’—I counted at least eight that could fit the description ‘person of color’ of one kind or another.

RESPONSE: This is also a good point. In two of the face databases we included for stimulus selection, race labels were provided (Chicago Face Database, London Face Database), therefore we filtered the faces based on those labels. In the other face database (Oslo Face Database), race labels were not provided; we aimed to eliminate obviously Asian or Black faces (for the race bias reasons noted above), but it is the case that even among the white faces, these are not all 100% Caucasian. We had all participants rate the faces with respect to how “white” they looked (with the definition that *“a person whose face looks like they are Caucasian”*), and this confirmed that while the faces were generally perceived to be white, there was some range on this attribution (Supplementary Fig.2a).

We have redone our analyses with excluding the faces whose “white” ratings were outliers ($n = 10$), and results with the remaining 90 faces (all with “white” ratings \geq the mid-point “4” of the rating scale) corroborated those found with the full set of faces: (1) the optimal number of factors to retain was four, as indicated by four methods together — Horn’s parallel analysis, Cattell’s scree test, optimal coordinates, and empirical BIC; (2) the four factors recovered from this subset was identical to those reported in the paper —Tucker’s indices of factor congruence = 1.00, 1.00, 1.00, 0.99 for the warmth, competence, femininity, youth dimensions between those from the subset and the full set of faces.

8.

Clarify that the four factors correspond to the actual words used in judgments, they are not the authors’ glosses of four abstract factors drawn from the model. I note with interest that the four factors all seem positively-valenced. Will the authors explain or speculate why these four traits seem important as discriminating factors? And are they implying that we only need these four words in order to most efficiently discriminate among face-trait judgements? What does it imply about how people distinguish or judge faces in real life?

RESPONSE: Actually this requires an important clarification. First, the four factors represent the relationships between groups of trait attributions from faces—some trait attributions are more strongly correlated with each other than with other trait attributions. We used ratings of the faces on the 100 trait words as input to the factor analyses, which quantify how the 100 trait attributions were correlated and grouped—in our case, they grouped into four factors (Supplementary Fig. 4a). The factor analyses themselves, including exploratory factor analysis (EFA), principal component analysis (PCA), and autoencoder artificial neural network (ANN), only quantify how the 100 trait attributions are grouped into the four factors, but they do not provide labels for the factors. We could have just labeled them “Factor 1”, “Factor 2”, “Factor 3”, and “Factor 4”. We put a single word label on each of the factors, as is commonly done in the literature, both to help

identify them consistently in the paper and to roughly indicate what social attribution concepts they describe. Therefore, the four labels are indeed our glosses of the four abstract factors drawn from the model. We have clarified this in the revised Abstract, emphasizing the four dimensions are “*interpreted as*” describing attributions of warmth, competence, femininity, and youth; in the Results, “*We interpreted these four factors as describing attributions of warmth, competence, femininity, and youth*” (page 9, line 1); we reiterate this in the Discussion (page 20, lines 5-6) and explain more on the labeling of the four dimensions in Methods, “*Dimensionality reduction methods do not provide labels for the factors discovered, which must instead be interpreted by the investigators....*” (page 33, lines 7-15).

That being said, we did verify that the label we subjectively determined indeed captured the relationship between the trait attributions as described by the factor. To do so, we collected data from an independent set of participants, where they were asked to evaluate, how “warm” the person described by each of the 100 trait words would be (N = 30 participants), and how “competent” the person described by each of the 100 trait words would be (N = 30 participants). We averaged these ratings across participants per trait word per label (“warm”, “competent”), and then computed the correlation between these ratings and the factor loadings across the 100 trait words. We found that the label “warmth” is most relevant for Factor 1 (absolute $r = 0.773$), and the label “competence” is most relevant for Factor 2 (absolute $r = 0.757$). For Factor 3 and Factor 4, we label them “femininity” and “youth” because those two trait attributions (feminine, youthful) had the highest factor loadings across all samples in Study 1 and Study 2 (Supplementary Fig. 4a, and Supplementary Fig. 6) and across all analysis methods (Supplementary Fig. 5a).

Note that the fact that a factor is not equivalent to any single trait attribution (but describes the relationships between a group of trait attributions) is also relevant for the linguistic problems noted earlier: it is quite possible that face ratings obtained in two languages, even with words for which no attempt is made at translation at all, nonetheless yield more or less the same four underlying factors. As mentioned before, the four factors we report here are not simply the semantic factors that describe the semantic relationship between the words themselves (page 26, lines 21-22; page 23, lines 1-13), but the factors that describe how the *faces* are represented.

Second, regarding the positive valence of the labels, in fact, each factor has two poles (Supplementary Fig. 4a): some trait attributions were positively correlated with the factor (red bars in the figure; e.g., attribution of critical from the face had a positive factor loading on Factor 1), some were negatively correlated with the factor (blue bars in the figure; e.g., attribution of easygoing from the face had a negative factor loading on Factor 1). Importantly, the sign of the factor loadings is arbitrary; that is, if we flip the sign of the factor loadings of all traits on Factor 1 (e.g., the sign of the loading of critical attribution becomes negative, and the sign of the loading of easygoing attribution becomes positive), Factor 1 remains unchanged — it still represents the same group of trait attributions that are more closely related to each other. Therefore, it is a common practice to name a factor following either pole of the factor; for example, we could have

named Factor 1 “not warm” or “cold”, etc. We choose the label “warmth” because, as mentioned before, it did capture the relationship between the trait attributions as described by Factor 1, and it is a dimension that has long been proposed in the larger social cognition literature.

Third, as for why these four dimensions turned out to be important as discriminating factors, it is helpful to situate our findings in the broader literature. Warmth and competence have long been proposed to be two universal dimensions of social cognition; for instance, the stereotype content model theorizes that these two dimensions capture the most variance in how people perceive different social groups (typically no face stimulus was used in this line of research). Our finding suggests that these two dimensions also seem to describe a large proportion of variance in how people perceive faces. The femininity and youth dimensions resonate with recent neuroimaging findings that social categorization (e.g., gender, age, race) shapes face perception. We mention these in the Discussion (page 20, lines 10-19).

Finally, we address the questions regarding implications of the four dimensions. In our data, these four dimensions accounted for 85% of the common variance in the ratings data. They show some connections with dimensions previously identified in the literature and seem to be the dimensions that are most useful to people when they need to represent other people from their faces. However, these four dimensions cannot be derived just from face ratings on four words (a larger number of variables than dimensions is needed for factor analysis)—in our paper we show what happens to the factors as the number of word ratings is decimated (Fig. 5a), and list a much smaller set of 18 words that we recommend for future research from which the four factors could be efficiently obtained (Supplementary Table 2b).

9.

Note that in evolutionary terms, this study tests an unusual task, because in the ancestral environment we would not have been confronted by unfamiliar faces very often; would the authors argue that these judgement tendencies would be adaptive in the ancestral human context?

RESPONSE: This is a very good question. In fact, there is a large literature that studies why people develop the tendency to rapidly attribute traits to others from faces in evolution. For instance, the overgeneralization theory (e.g., Zebrowitz & Montepare, 2008) argues that some facial cues such as those that signal low fitness, babies, emotion were so useful for our ancestors that we overgeneralize them: we attribute low fitness (e.g., “unhealthy”) to individuals whose faces look atypical; we attribute the qualities of babies (e.g., “warm”, “submissive”) to individuals with a babyface; and we attribute the qualities of people who are having different emotions (e.g., people who are angry are likely to do harm) to individuals whose (neutral) face resembles certain emotions (e.g., anger). That is, these theories argue that the association between certain facial cues and individual qualities that were adaptive for our ancestors became overgeneralized and applied to a much wider set of facial cues (e.g., from cues of a baby’s face, to cues of baby-faced individuals’ faces) – one likely source of our

stereotypes and biases. We have now mentioned this literature in our revised Discussion (page 21, lines 22-23; page 22, lines 1-7).

In addition, related to our responses to the reviewer's previous comments, at least two aspects would contribute importantly to trait attributions from faces in real life and in our ancestral environment, and were omitted here: (1) context (we usually see people doing things, not faces in isolation), and (2) familiarity (we usually encounter faces over and over again and can link them to familiar individuals). We elaborate on these in our revised Discussion (page 23, lines 14-18).

10.

The discussion falls flat, by merely gesturing toward future work that would address shortcomings of this study. But there are interesting implications and they should be addressed. For example: what is the relation between people's trait judgements and the real traits of people? There is a mismatch in social discourse around human traits: on the one hand, people appear to consistently judge certain faces as appearing 'competent', but on the other hand people want to say that it is impossible to tell from somebody's face whether they are actually a competent person. How can this research help counteract the human propensity to judge people by their faces? The study should finish with some straightforward questions to be asked next, not a request that subsequent studies should address shortcomings of this one.

RESPONSE: We agree that questions like these are interesting; in fact, each of them has spawned an entire literature. For example, the relation between people's trait attributions from faces and the real traits of people has been investigated by a large literature, and the conclusion remains debated: some studies report evidence that the two are correlated (e.g., a recent response from Bonnefon, Hopfensitz, & Neys, 2015; Penton-Voak, Pound, Little, & Perrett, 2006, about attributions of personality traits from faces and actual personality; Valla, Ceci, & Williams, 2011, about attributions of criminality from faces and actual criminal records), others report evidence that trait attributions from faces are mostly inaccurate (e.g., Todorov, 2017; Rule, Krendl, Ivcevic, & Ambady, 2013). A critical challenge to this line of research is how one should define the "ground truth" (e.g., of whether somebody is competent): on the one hand, it is subject to particular situations and opinions (some of their actions, in certain situations, or in the opinions of some other people, might look competent; some not); on the other hand, it is easily confounded with other types of human judgments that are also subject to the same biases (e.g., a politician whose face looks competent getting elected to office does not necessarily show that competence attribution from faces is accurate, since voters' decisions might also have been biased by how competent the politician looks). In most cases, there is no clear operationalization of what would count as the "ground truth" for most traits that we attribute to people—consensus judgments by other people (or perhaps a specialized group of "experts") is often all we have (e.g., averaged peer-reported personality traits of an individual). We raise this deep conceptual issue in our Discussion now as well—it is not clear that there is any objective criterion to determine people's traits as commonly conceived (page 21, lines 17-22).

There is also a considerable research on the mismatch between people making attributions from faces that show high consensus and these attributions not being valid. For instance, the overgeneralization theory we mentioned previously (e.g., Zebrowitz & Montepare, 2008) provides a plausible explanation for why people make such attributions from faces at the first place, why these attributions tend to have high consensus across perceivers, and how they might have gone wrong over time. We allude to such question and related literature in our Discussion now as well (page 21, line 22-23; page 22, lines 1-7).

Our present study could not draw any conclusion regarding the validity of these trait attributions, since the actual traits of the people whose faces were shown in the study are unknown. Our study does not directly provide any remedy to counteract the human propensity to attribute traits to others based on faces either; however, our study does point out the dimensions that summarize the biases and stereotypes of these trait attributions, which could be targeted for remedy. We have now added this to the Impact Statement, *“Indeed, our findings identify four important dimensions that may contribute to biases and stereotypes that people exhibit when viewing faces, informing future work on stereotyping.”* The many factors omitted in our study immediately lead to the questions to be asked next: how language might modify the psychological dimensions of trait attributions from faces; how race, age, and other social categorization of the face might modify these dimensions, etc. We have now clarified these future directions in the Discussion (page 21, lines 4-5; page 23, lines 3-10; page 24, lines 2-4).

11.

The discussion should address explanations: WHY would people have strong and consistent judgments about whether a stranger is ‘abusive’ or ‘conscientious’ based on their face? Are these judgments hardwired? Are the judgments quickly overwritten when we get to know the person? Or is it just that we are wired with a drive to classify faces in terms of traits and the actual mapping of faces to traits is a result of historical contingencies, eg, tropes based on movies (look at Bond villains, horror films, etc. over the decades)? Are the mappings learned? These are crucial questions and the study doesn’t address them.

RESPONSE: We agree with the reviewer that these are interesting and difficult questions. However, our study cannot answer them, and we do not wish to speculate. There is a large body of research that studies why people make these trait attributions from faces (e.g., the overgeneralization theory), but it is still under debate. Regarding the question about how quickly these trait attributions could be overwritten when we get more information about the person, there is also substantial research, but again it is still under debate (e.g., Shen, Mann, & Ferguson, 2020; Rezlescu, Duchaine, Olivola, & Chater, 2012). On the stimulus end, things are similarly murky: aside from some links between sex steroids and face width-to-height ratio, most of the geometric structure of the face develops from unknown myriad factors. How these might in any way be associated with that person’s traits is unknown (and, as noted above, it is unlikely that the traits for which we have concepts in English, or presumably any other language, correspond to any biological natural category –“competence”, for instance, is essentially

an observer-relative judgment).

12.

Line 326: 'additional research will be needed to elucidate how language might modify the psychological space of face judgments'. A relevant reference to note is the finding that describing a face in words causes the speaker to be worse at remembering and later identifying the face (by a process of 'verbal overshadowing'):

Schooler, Jonathan W, and Tonya Y Engstler-Schooler. 1990. "Verbal Overshadowing of Visual Memories: Some Things Are Better Left Unsaid." *Cognitive Psychology* 22 (1): 36–71. [https://doi.org/10.1016/0010-0285\(90\)90003-M](https://doi.org/10.1016/0010-0285(90)90003-M).

Alogna, V. K., M. K. Attaya, P. Aucoin, Š. Bahník, S. Birch, A. R. Birt, B. H. Bornstein, et al. 2014. "Registered Replication Report: Schooler and Engstler-Schooler (1990)." *Perspectives on Psychological Science* 9 (5): 556–78. <https://doi.org/10.1177/1745691614545653>.

RESPONSE: We thank the reviewer for these interesting references and have included them in our revised paper (page 23, line 10).

Reviewer #4 (Remarks to the Author):

The authors have an ambitious and highly interesting research aim which uses word reduction techniques to generate a small set of non-redundant trait words, approaching the issue of impression formation from a different angle than previous work, which has focused on spontaneously attributed impressions of faces. In many ways, the work is reminiscent of a modern take on Allport's original, classic studies into trait impressions in the 30s (Allport & Odbert, 1936), albeit with up-to-date statistical analyses and tools which were not available in Allport's day. To be clear, I am in favour of publication and think the aims and methodological rigour of the work alone makes it worth publication. My comments are largely aimed at addressing the theoretical contribution, and in particular I would like the authors to acknowledge existing theory in more depth.

RESPONSE: We thank the reviewer for the positive assessment and the helpful comments below. In particular, we have performed new analyses, and confirmed that, while our data showed weak correspondence between the first two dimensions reported here and the valence and dominance dimensions previously proposed, the youth dimension we reported here was in fact highly correlated with the youthful/attractiveness found in the reviewers' previous study ($r = 0.71$ based on EFA scores, $r = 0.76$ based on PCA scores). Please see our response in detail below.

First, are the dimensions of warmth and competence really that different from valence and dominance? I wasn't convinced that the differences were as large as stated and the authors risk overextending their interpretation here. The Stereotype Content Model also shares theoretical similarities with Alex Todorov's model which go beyond the specific

labels of warmth etc; i.e. both suggest that at the core, trait judgements are based on approach/avoid orientation. Particularly, I would urge the authors to consult previous studies which have also investigated the question how warmth and competence relate to trustworthiness and dominance and/or across populations of faces; Walker et al 2017, JPSP, Sutherland et al 2016, Cognition, 2018, PSPB Collova et al 2019, JPSP. Some of these studies find greater similarity than others, and particularly, dominance and competence appear to diverge more for populations other than White males (i.e. dominance appears to be less important for characterising female faces, Asian faces, and children's faces). The authors did a good job of acknowledging the limitations of their face stimuli in the abstract/discussion, but I would like to see greater discussion of this point which includes this previous literature. The authors could also consider the question of whether trait space is actually more flexible than previously recognised (Stolier et al, 2018, TICS).

RESPONSE: These are excellent suggestions. We have now acknowledged these previous studies that surveyed the relationship between the warmth/competence and trustworthiness/dominance frameworks in the Discussion, *"Our findings add to those from previous studies^{64,65} that attempted to reconcile the dimensions of face perception with the dimensions from the broader social cognition literature..."* (page 20, line 13). We agree that the meaning of the dimensions is not bound rigidly to the meaning of the specific words used as labels. We qualify our interpretation of the extent to which the dimensions we report here are the same or differ from those previously reported. In our data, the first dimension showed moderate to weak correlation with the previously found valence dimension, and the result varied largely depending on the analysis method ($r = 0.41$ based on EFA scores; $r = 0.09$ based on PCA scores); the second dimension was not significantly correlated with the previously found dominance dimension ($r = 0.01$ based on EFA scores; $r = 0.09$ based on PCA scores). However, as pointed out by the reviewer, we agree that the relationships between these different dimensions are likely to vary depending on what type of face stimuli were used, which we have also acknowledged in the revised Discussion, *"However, the relationships between these different dimensions are likely to be modified by the stimuli used^{2,23} (see below)." (page 20, lines 12-13).*

We have expanded our discussion of the limitations of our face stimuli and cited the papers the reviewer suggests (page 22, lines 8-21; page 23, lines 11-21), and indeed now note the possibility that trait space may be quite flexible (page 23, lines 3-4).

Similarly, it looked like the "youth" factor in the 4D solution appeared to replicate the "youthful-attractiveness" factor found by Sutherland et al. (2013), but the authors could clarify here. Presumably there is a strong conceptual similarity if both rely on age, but where does attractiveness sit in the four dimensions? Attractiveness looked to cross-load in the three dimensional replication, perhaps an indication that age is the more robust or distinctive aspect of this dimension? (also perhaps why this third dimension appeared in our 2013 study with more variable ambient images that differed on age, but not in Alex Todorov's original work, where age was not varied).

RSEPONSE: The reviewer is correct. We have now quantified the relation between the third dimension we found here and the youthful/attractiveness dimension found by Sutherland et al. (2013): the two were indeed highly correlated ($r = 0.71$ based on EFA scores; $r = 0.76$ based on PCA scores). We have now reported this in the Results (page 14, lines 2-4), and emphasized again in the Discussion, *“The youth dimension we found here resonates with the youthful/attractiveness dimension found in prior work that used more diverse face images that differed on age³⁶”* (page 20, lines 16-18). As for where attractiveness sits in the four dimensions, it is not straightforward: the trait word “attractiveness” was replaced by “beautiful” in the word sampling process based on meaning clarity and usage frequency; “beautiful” indeed cross-loaded in the three dimensional replication, and did not consistently load on one dimension (in most of the samples it was most strongly associated with the competence dimension, and in others with the youth dimension; see Supplementary Fig. 4a, and Supplementary Fig. 6). Indeed, in our data, age was found to be the more robust and distinctive aspect of the fourth dimension; in particular, when we reanalyzed our data using PCA and forced the factors to be orthogonal with each other, “youthful” was the only trait attribution that loaded on the fourth dimension (Supplementary Fig. 5a). We agree with the reviewer that the age variance in the face stimuli is plausibly very important for this dimension to emerge.

Second, I am very reassured that the authors replicate their work with multiple modelling methods, including EFA with oblimin and PCA, as well as a new nonlinear technique. This paper will have an important contribution to make given the current debate over the best approach for dimensionality reduction across culture (e.g. Jones et al Nature Human Behaviour, in press, Oh and Todorov, in press). We have followed a similar logic to assessing whether models are robust and similarly did not find large differences between analyses. I also appreciated the replication with reference to previous published models, as this approach was very helpful in outlining similarities and differences across studies.

RESPONSE: We thank the reviewer for the positive comments. We have also included these references in our revised manuscript (page 11, line 16).

Finally, it is also reassuring that the authors broadly replicated their models across culture. The cross-cultural work speaks to a current question in the literature as to how universal these trait impressions are (Jones et al Nature Human Behaviour, in press, Walker et al 2011, SPPS, Sutherland et al 2018, PSPB, Zebrowitz et al J Cross Cultural Psychology, 2012). The work of Mirella Walker and her colleagues should again be noted here; her team was arguably the first to test face trait models across culture. Leslie Zebrowitz also carried out outstanding work into impressions from faces made by non-Western cultures, a research effort which I would also like the field to acknowledge more clearly when culture is investigated.

RESPONSE: We thank the reviewer for these positive comments and for suggesting these relevant works. We have now included these references to our revised manuscript (page 16, line 1). We note that we have also incorporated the concerns of

reviewer 3, who pointed out the issue with language in operationalizing cross-culture studies (e.g., participants from other regions whose native language is not English might understand our trait words differently, and other languages like Spanish do not have direct translational equivalents of our words) — we note these limitations and clarify that we refrain from drawing any specific conclusions about cultural differences or universality (page 21, lines 9-13).

Clare Sutherland

Minor issues: The authors are to be commended for investigating this question at the individual participant level, as it involves many hours of testing to get a full set of data from one participant; moreover, they managed to get samples across culture here too. We (Sutherland et al 2020, BJP) also managed to reproduce a particular (3D) model at the individual level in a handful of participants with a smaller set of traits, but the current paper goes far beyond what we attempted. Really nice work.

RESPONSE: Thank you. We have added a reference to your prior work now as well (page 18, line 11; page 21, line 10).

Could the focus on sampling words (as is typically carried out in social cognition research), rather than faces (as is typically carried out in face perception studies), have led to potential differences across trait and face models i.e. warmth v trust? I think not, given the supplementary analyses, but the authors may wish to specify, as this question was previously unclear e.g. Sutherland et al 2016, Cognition.

RESPONSE: We thank the reviewer for this observation. Indeed, our supplemental analysis, which decimated the words, still found considerably robust recovery of the four dimensions. However, our study cannot offer a definite answer to this question: one would need to have self-report or peer-report ratings on these words (without showing any face) and perform the word decimating analysis to see whether the warmth dimension remains robust or it gets replaced by the trust dimension when fewer and fewer words are left. We think the sampling probability of representatively sampled words in typical face perception studies may have contributed to the difference across the trait and face models. We suspect that, with a representatively sampled set of faces, even without giving any words but just asking participants to rate the pair-wise similarity between the faces based on instructions such as “how similar these people look in terms of what type of person they are”, it would be possible to still recover these four dimensions, or dimensions that are highly similar to those from social cognition research.

Carryover effects (Rhodes et al 2006). Participants rated multiple traits, so I did wonder if this procedure could affect the results. Is it possible to examine the trait space with just the first rating given by participants? Does it change anything?

RESPONSE: This is an interesting point. Since the order of traits was randomized across participants, whatever carryover effects there may have been would be expected

to wash out. But the point is well-taken and we have performed the analysis the reviewer suggests: for each trait, we only included ratings from participants who rated this trait in the first block (the number of remaining participants ranged from 12 to 16 per trait); using this subset of data, we reproduced the four dimensions found in the full set of data: the Tucker indices of factor congruence between the four dimensions found in the subset and the full set were 0.98, 0.97, 0.93, 0.92. We have now reported this in the Methods (page 31, lines 2-4).

Line 622 - Hehman et al (2017, JPSP) also showed that traits relating to physical aspects showed higher between-subjects consistency than more abstract traits.

RESPONSE: Thanks for this reference; we have added the citation to this part (page 37, line 22).

I thought it was a service to the field that the authors recommend a subset of 18 words which could be used as a shorthand, but it would also be useful to know which words absolutely had the best measurement invariance (i.e. what if studies can only recruit ten words, or even only four words?)

RESPONSE: This is a helpful question. If measurement precision (for other purposes than a factor analysis) is desired, then the within-subject test-retest reliability and the between-subject consensus that we quantify for the trait attributions is probably the best metric to use. We have clarified this in our paper (page 15, lines 12-14), *“For studies with more stringent constraints on the number of trait words (e.g., due to limited testing time available), an even smaller subset may be selected based on the within-subject consistency and between-subject consensus (Fig. 3; e.g., easygoing, competent, femininity, youthful).”* However, we would not recommend factor analyses based on such a small number of trait attributions.

I'm not sure I followed Fig 7. Which dimensions are which in this figure? Is dimension 3, “femininity”, and dimension 4, “youth”? Or are they different across cultures?

RESPONSE: Fig. 7c only shows the number of dimensions according to parallel analysis (our preregistered method), without presenting the interpretation of those dimensions; we have now clarified this in the figure legend (page 20, lines 1-2). Regarding the interpretation of the dimensions, we refer to the actual factor loadings and the Tucker index coefficients (Supplementary Figure 7): as shown in those figures, not all participants whose data showed a four-dimensional structure (in Fig. 7c) exactly replicated the four dimensions found in Study 1; only for a handful of participants we could say that their 3rd dimension described femininity, and their 4th dimension described youth.

Line 103 ICCs (and in the methods) - which ICCs specifically? The authors could give more justification and relate to a specific case model from McGraw and Wong (1996). I would prefer a two-way random model - subjects in columns should also be random for example (see p37). i.e. a two-way random model (C, 1), is the closest approximation to

alpha at the individual level. See also p38. The authors may indeed have used this ICC so it may just be a case of clarification.

RESPONSE: Yes, we used the ICC for two-way random-effects models for consistency; we used R function *ICC(2,k)* (in the “psych” package) to compute the coefficients, which provided options for 6 ICCs from Shrout and Fleiss (1979); we have clarified this in the revised manuscript (page 32, line 5).

REVIEWER COMMENTS

Reviewer #1 (Remarks to the Author):

The authors have been responsive to reviewers' concerns. I was happy with a previous version of the paper. After reading this revised version I see no reason to change my decision. I believe this paper presents an interesting contribution to the literature.

Reviewer #2 (Remarks to the Author):

I still do not have remaining issues. The last rounds of revision improved the manuscript. I recommend publication.

Reviewer #3 (Remarks to the Author):

I am satisfied that the authors have carefully reviewed and responded to my earlier suggestions. Based on their response, I would be happy to see the article published.

N. J. Enfield

Reviewer #4 (Remarks to the Author):

I read the authors' response to all reviews carefully and was very impressed with the measured and thorough response. I have no further comments - the authors are to be congratulated on a very impressive paper.

Clare Sutherland

I'm not sure I followed Fig 7. Which dimensions are which in this figure? Is dimension 3, "femininity", and dimension 4, "youth"? Or are they different across cultures?

Line 103 ICCs (and in the methods) - which ICCs specifically? The authors could give more justification and relate to a specific case model from McGraw and Wong (1996). I would prefer a two-way random model - subjects in columns should also be random for example (see p37). i.e. a two-way random model (C, 1), is the closest approximation to alpha at the individual level. See also p38. The authors may indeed have used this ICC so it may just be a case of clarification.

REVIEWERS' COMMENTS

Reviewer #1 (Remarks to the Author):

The authors have been responsive to reviewers' concerns. I was happy with a previous version of the paper. After reading this revised version I see no reason to change my decision. I believe this paper presents an interesting contribution to the literature.

RESPONSE: We thank the reviewer for their positive assessment.

Reviewer #2 (Remarks to the Author):

I still do not have remaining issues. The last rounds of revision improved the manuscript. I recommend publication.

RESPONSE: We thank the reviewer for their positive assessment.

Reviewer #3 (Remarks to the Author):

I am satisfied that the authors have carefully reviewed and responded to my earlier suggestions. Based on their response, I would be happy to see the article published.

N. J. Enfield

RESPONSE: We thank the reviewer for their positive assessment.

Reviewer #4 (Remarks to the Author):

I read the authors' response to all reviews carefully and was very impressed with the measured and thorough response. I have no further comments - the authors are to be congratulated on a very impressive paper.

Clare Sutherland

RESPONSE: We thank the reviewer for their positive assessment.